# PDE-GAN FOR SOLVING PDEs OPTIMAL CONTROL PROBLEMS MORE ACCURATELY AND EFFICIENTLY

## Abstract

PDEs optimal control (PDEOC) problems aim to optimize the performance of physical systems constrained by partial differential equations (PDEs) to achieve desired characteristics. Such problems frequently appear in scientific discoveries and are of huge engineering importance. Physics-informed neural networks (PINNs) are recently proposed to solve PDEOC problems, but it may fail to balance the different competing loss terms in such problems. Our work proposes PDE-GAN, a novel approach that puts PINNs in the framework of generative adversarial networks (GANs) "learn the loss function" to address the trade-off between the different competing loss terms effectively. We conducted detailed and comprehensive experiments to compare PDEs-GANs with vanilla PINNs in solving four typical and representative PDEOC problems, namely, (1) boundary control on Laplace Equation, (2) time-dependent distributed control on Inviscous Burgers' Equation, (3) initial value control on Burgers' Equation with Viscosity, and (4) time-space-dependent distributed control on Burgers' Equation with Viscosity. Strong numerical evidence supports the PDE-GAN that it achieves the the best control performance and shortest computation time without the need of line search which is necessary for vanilla PINNs.

## 1 Introduction

In physics, partial differential equations (PDEs) hold significant scientific and engineering importance. Controlling the behavior of systems constrained by PDEs is crucial for many engineering and scientific disciplines (Chakrabarty & Hanson, 2005). PDEs optimal control (PDEOC) problems aim to optimize the performance of physical systems governed by PDEs to achieve desired characteristics (Lions, 1971). The standard mathematical expression of the PDEOC problem is as follows.

Consider a physical system defined over a domain $\Omega \subset \mathbb{R}^d$, governed by the following PDEs and cost objectives.

$$\min_{\mathbf{u} \in U, \mathbf{c} \in Y} \mathcal{J}(\mathbf{u}, \mathbf{c}), \quad \text{subject to} 1b, 1c, 1d \tag{1a}$$

$$\mathcal{F}[\mathbf{u}(\mathbf{x}, t), \mathbf{c}_v(\mathbf{x}, t)] = 0, \quad \mathbf{x} \in \Omega, \, t \in [0, T], \tag{1b}$$

$$\mathcal{B}[\mathbf{u}(\mathbf{x}, t), \mathbf{c}_b(\mathbf{x}, t)] = 0, \quad \mathbf{x} \in \partial\Omega, \, t \in [0, T], \tag{1c}$$

$$\mathcal{I}[\mathbf{u}(\mathbf{x}, 0), \mathbf{c}_0(\mathbf{x})] = 0, \quad \mathbf{x} \in \Omega, \, t = T. \tag{1d}$$

Here, $\mathbf{x}$ and t denote the spatial and temporal variables. $\mathcal{J}(\mathbf{u}, \mathbf{c})$ represents the cost objective to be minimized and $\mathbf{c} = (\mathbf{c}_v, \mathbf{c}_b, \mathbf{c}_0)$, which correspond to distributed control, boundary control, and initial value control, respectively. The terms $\mathcal{F}$, $\mathcal{B}$ and $\mathcal{I}$ represent the constraints that the system state $\mathbf{u}$ and the optimal control $\mathbf{c}$ must satisfy, which encompass the PDE residual, as well as the boundary and initial conditions. $U$ and $Y$ denote the appropriate spaces where $\mathbf{u}$ and $\mathbf{c}$ belong to.

So far, various methods have been developed to solve PDEOC problems. Recently, deep learning-based solving methods using PINNs (Physics-Informed Neural Networks) have

gained widespread attention. Raissi et al. (2019) introduced the concept of PINNs in 2017, which fundamentally transformed the traditional and uninterpretable approach of training neural networks solely based on large amounts of observational data like a black-box. In the framework of PINNs, the system state $\mathbf{u}(x,t)$ is represented by a surrogate model $\mathbf{u}_{\theta_u}(x,t)$ in the form of a fully-connected neural network, where $\theta_u$ denotes the set of trainable parameters of the network. For prescribed control variables $\mathbf{c} = (\mathbf{c}_v, \mathbf{c}_b, \mathbf{c}_0)$, the network parameters $\theta_u$ are trained by minimizing the loss function (2a).

$$\mathcal{L}(\theta_u) = \mathcal{L}_{\mathcal{F}}(\mathbf{u}_{\theta_u}, \mathbf{c}_v) + \mathcal{L}_{\mathcal{B}}(\mathbf{u}_{\theta_u}, \mathbf{c}_b) + \mathcal{L}_{\mathcal{I}}(\mathbf{u}_{\theta_u}, \mathbf{c}_0), \tag{2a}$$

$$\mathcal{L}_{\mathcal{F}}(\mathbf{u}_{\theta_u}, \mathbf{c}_v) = \frac{1}{N_f} \sum_{i=1}^{N_f} \left| \mathcal{F}[\mathbf{u}_{\theta_u}(x_i^f, t_i^f), \mathbf{c}_v] \right|^2, \tag{2b}$$

$$\mathcal{L}_{\mathcal{B}}(\mathbf{u}_{\theta_u}, \mathbf{c}_b) = \frac{1}{N_b} \sum_{i=1}^{N_b} \left| \mathcal{B}[\mathbf{u}_{\theta_u}(x_i^b, t_i^b), \mathbf{c}_b] \right|^2, \tag{2c}$$

$$\mathcal{L}_{\mathcal{I}}(\mathbf{u}_{\theta_u}, \mathbf{c}_0) = \frac{1}{N_0} \sum_{i=1}^{N_0} \left| \mathcal{I}[\mathbf{u}_{\theta_u}(x_i^0, 0), \mathbf{c}_0] \right|^2, \tag{2d}$$

where $\{(x_i^f, t_i^f)\}_{i=1}^{N_f}$, $\{(x_i^b, t_i^b)\}_{i=1}^{N_b}$, $\{(x_i^0, 0)\}_{i=1}^{N_0}$ each represent an arbitrary number of training points over which to enforce the PDE residual (1b), boundary conditions (1c), and initial condition (1d), respectively. In addition, $\mathcal{L}_{\mathcal{F}}$, $\mathcal{L}_{\mathcal{B}}$ and $\mathcal{L}_{\mathcal{I}}$ are referred to as the PDE loss, boundary loss, and initial value loss, respectively.

Recently, Mowlavi & Nabi (2023) investigated ways to utilize PINNs to solve PDEOC problems. In their works, they used distributed control as an example ($\mathbf{c} = \mathbf{c}_v$) to illustrate how to extend PINNs to solve optimal control problems. They introduced a second fully-connected neural network $\mathbf{c}_{\theta_c}(x,t)$ to find the optimal control function $\mathbf{c}$. PINNs are learnt by enforcing the governing equations at the points in the domain and its boundary. The core idea is to incorporate the cost objective ($\mathcal{J}$) into the loss (2a) to construct the augmented loss function (3). Boundary and initial value control are similar to the above.

$$\mathcal{L}(\theta_u, \theta_c) = \mathcal{L}_{\mathcal{F}}(\mathbf{u}_{\theta_u}, \mathbf{c}_{\theta_c}) + \mathcal{L}_{\mathcal{B}}(\mathbf{u}_{\theta_u}) + \mathcal{L}_{\mathcal{I}}(\mathbf{u}_{\theta_u}) + \omega \mathcal{L}_{\mathcal{J}}(\mathbf{u}_{\theta_u}, \mathbf{c}_{\theta_c}), \tag{3}$$

$$\mathcal{L}_{\mathcal{J}}(\mathbf{u}_{\theta_u}, \mathbf{c}_{\theta_c}) = \mathcal{J}(\mathbf{u}_{\theta_u}, \mathbf{c}_{\theta_c}), \tag{4}$$

where $\mathcal{L}_{\mathcal{J}}(\mathbf{u}_{\theta_u}, \mathbf{c}_{\theta_c})$ is denoted as Cost loss, $\omega$ denote the cost objective weight and the subscripts on $\mathbf{u}$ and $\mathbf{c}$ indicate the dependence to neural network parameters $\theta_u$ and $\theta_c$.

Their works address the trade-off between the different competing loss terms in PDEOC problems through line search on the cost objective weights ($\omega$). The vanilla PINNs make use of a two-step line search method to identify the optimal weight $w$. In details, the two-step method trains a separate pair of solution and control networks for each weight, and then predict their corresponding final states. Such final states will be compared with the analytical solution, and the $w$ attaining the least error is then chosen. In other words, such method searches for the optimal weight $w$ exhaustively. One of the obvious drawbacks is thus its heavy cost of computation time. Also, such line search method serves only as a heuristic, lacking strong theoretical support, which makes further analysis in robustness and stability challenging. Therefore, it is imperative to develop an effective strategy to handle PDEs constraints for solving PDEOC problems.

To address this theoretical gap, we put PINNs in the framework of Generative Adversarial Networks (GANs) (Goodfellow et al., 2020) to solve PDEOC problems in a fully unsupervised manner. Inspired by Zeng et al. (2022), we adaptively modify the entire loss function throughout the training process, rather than just changing the weights of the loss terms, to improve the accuracy of the solution. Our PDE-GAN uses the discriminator network to optimize the generator's loss function, eliminating the need for predefined weights and offering greater flexibility compared to line search methods.

Our contributions in this work are summarized as follows:

- We propose a novel approach for solving PDEs optimal control problems, namely, PDEs-GANs, which is capable of "learning the loss function" in the learning process.

- Our method, PDEs-GANs, is the first to incorporate PINNs into the framework of GANs to solve PDEs optimal control problems, with the benefit of balancing different competing loss terms much more efficiently and effectively.

- Our method, PDEs-GANs, can provide more accurate solutions in less computation time than vanilla PINNs, as demonstrated in our numerical experiments on optimal control problems of Laplace equation and Burgers' equation.

The remainder of this paper is structured as follows. Section 2 introduces related work on solving PDEOC problems. Section 3 presents our method PDE-GAN for solving PDEOC problems. Section 4 describes our empirical studies and then discusses the effectiveness of our method compared to hard-constrained line search method and Soft-PINNs line search method. Section 5 concludes our findings.

## 2    Related work

Various methods have been developed for solving PDEOC problems, which can be mainly divided into traditional numerical method and deep-learning based approaches.

The adjoint method (Herzog & Kunisch, 2010), as one of the traditional approaches for solving PDEOC problems, has been successfully applied to optics and photonics (Bayati et al., 2020; Molesky et al., 2018; Pestourie et al., 2018), fluid dynamics (Borrvall & Petersson, 2003; Duan et al., 2016), and solid mechanics (Bendsoe & Sigmund, 2013; Sigmund & Maute, 2013). It is based on Lagrange's famous 1853 paper (Lagrange, 1853), which laid the foundation for Lagrange multipliers and adjoint-based sensitivity analysis. This method involves iteratively computing the gradient of the cost objective with respect to optimal control solutions until stopping conditions are met. It works by solving a second adjoint PDEs equation in addition to the original control equation.

Although the adjoint method is a powerful tool for solving PDEOC problems, it has significant drawbacks. First, deriving the adjoint PDEs equations for simple optimal control problems with complex PDEs is a challenging task. Moreover, the adjoint method relies on finite element or finite difference methods, and its computational cost increases quadratically to cubically with the mesh size. Therefore, solving PDEOC problems with large search spaces and mesh sizes becomes extremely expensive and may even become intractable, which is known as the curse of dimensionality.

To resolve those problems, various deep-learning based methods have been developed for solving PDEOC problems. Some of these are supervised, such as Lu et al. (2019), where the authors use DeepONet to replace finite element methods. They use DeepONet to directly learn the mapping from optimal control solutions to PDEs solutions and further replace network constraints with PDEs networks. However, these methods require pre-training a large operator network, which is both complex and inefficient. Moreover, if the optimal solution lies outside the training distribution, performance may degrade (Lanthaler et al., 2022).

To improve training accuracy, Demo et al. (2023) utilized physical information in various ways. They used physical information as enhanced input (additional features) and as a guide for constructing neural network architectures. This approach accelerated the training process and improved the accuracy of parameter predictions. However, it remains to be verified which type of physical information is most suitable for use as enhanced input.

There is also an unsupervised neural network approach. For example, as we mentioned before, Mowlavi & Nabi (2023) proposed using a single PINN to solve PDEOC problems. This method introduces a trade-off between the cost objective and different competing loss terms, which is crucial for performance (Nandwani et al., 2019).

To resolve the trade-off between the different loss terms, Hao et al. (2022) formulated the PDEOC problem as a bi-level loop problem. They used implicit function theorem (IFT) differentiation to compute the hypergradient of the control parameters $\theta$ in the outer loop. In the inner loop, they fine-tuned the PINN using only the PDEs loss.

Although the bi-level method splits different competing loss terms, it creates an extra problem about the computation of hypergradient, the accuracy of which largely depends on the specific numerical methods applied. Therefore, applying the bi-level methods do not solve the trade-off problem directly but actually transform it to another pair of problems in solving hypergradient and PINN solution at the same time.

## 3 PDE-GAN

In this section, we introduce our method, PDE-GAN, which integrates PINNs into the GAN framework. Through the generative-adversarial interplay between the generator network and the discriminator network, the loss function is continuously optimized to learn the weights between the cost objective and the different competing loss terms in PDEOC problems.

### 3.1 Generative Adversarial Networks

Generative Adversarial Networks (GANs) (Goodfellow et al., 2020) are generative models that use two neural networks to induce a generative distribution $p(x)$ of the data by formulating the inference problem as a two-player, zero-sum game.

The generative model first samples a latent random variable $z \sim \mathcal{N}(0, 1)$, which is used as input into the generator $G$ (e.g., a neural network). A discriminator $D$ is trained to classify whether its input was sampled from the generator (i.e., "generated data") or from a reference data set (i.e., "real data").

Informally, the process of training GANs proceeds by optimizing a minimax objective over the generator and discriminator such that the generator attempts to trick the discriminator to classify "generated data" samples as "real data". Formally, one optimizes

$$\min_G \max_D V(D, G) = \min_G \max_D \Big( \mathbb{E}_{x \sim p_{\text{data}}(x)}[\ln D(x)] + \mathbb{E}_{z \sim p_z(z)}[1 - \ln D(G(z))] \Big),$$

where $x \sim p_{\text{data}}(x)$ denotes samples from the empirical data distribution, and $p_z \sim \mathcal{N}(0, 1)$ samples in latent space. In practice, the optimization alternates between gradient ascent and descent steps for $D$ and $G$ respectively.

### 3.2 Hard-Constrained Physics-Informed Neural Networks

In the Introduction, we presented the construction method of the loss function for solving PDEOC problems based on soft-constrained PINNs (Equations (3) and (4)). During the optimization process, the four loss terms—PDE residual condition, boundary condition, initial condition, and cost objective—compete for gradients, making the training results highly dependent on the choice of weights $\omega$.

To mitigate this issue, another PINNs-based method employs function transformations or neural network numerical embeddings to explicitly enforce the initial and boundary conditions on the surrogate system state neural network model $u_{\theta_u}(x, t)$. This reformulation reduces the four loss terms to just the PDE residual term and the cost objective term, significantly improving the performance of solving PDEOC problems.

Clearly, adjusting the weight relationship between two loss terms is more effective than adjusting four terms. To ensure the exact satisfaction of initial and boundary conditions, various methods can be employed. For instance, the neural network output $\mathbf{u}_{\theta_u}(x, t)$ can be modified to meet the initial condition $\mathbf{u}(x, t)|_{t=t_0} := \mathbf{u}_0$ (Lagaris et al., 1998). Ones can apply the re-parameterization :

$$\hat{\mathbf{u}}_{\theta_u}(x, t) = \mathbf{u}_0 + t\mathbf{u}_{\theta_u}(x, t), \tag{5}$$

which exactly satisfies the initial condition. Flamant et al. (2020) proposed an augmented re-parameterization

$$\hat{\mathbf{u}}_{\theta_u}(x, t) = \Phi(\mathbf{u}_{\theta_u}(x, t)) = \mathbf{u}_0 + (1 - e^{-(t-t_0)})\mathbf{u}_{\theta_u}(x, t), \tag{6}$$

that further improved training convergence. Intuitively, equation (6) adjusts the output of the neural network $\mathbf{u}_{\theta_u}(x,t)$ to be exactly $\mathbf{u}_0$ when $t = t_0$, and decays this constraint exponentially in $t$.

Therefore, the core idea of this method is to incorporate these reparameterized states $\hat{\mathbf{u}}_{\theta_u}$ into the augmented loss function 3 to construct a new augmented loss function (7):

$$\mathcal{L}(\hat{\mathbf{u}}_{\theta_u}, \mathbf{c}_{\theta_c}) = \mathcal{L}_{\mathcal{F}}(\hat{\mathbf{u}}_{\theta_u}, \mathbf{c}_{\theta_c}) + \omega \mathcal{L}_{\mathcal{J}}(\hat{\mathbf{u}}_{\theta_u}, \mathbf{c}_{\theta_c}) \tag{7a}$$

$$= \frac{1}{N_f} \sum_{i=1}^{N_f} \left| \mathcal{F}[\hat{\mathbf{u}}_{\theta_u}(x_i^f, t_i^f), \mathbf{c}_{\theta_c}] \right|^2 + \omega \mathcal{J}(\hat{\mathbf{u}}_{\theta_u}, \mathbf{c}_{\theta_c}) \tag{7b}$$

It can be seen that, unlike the loss function under equation (3), the loss term in equation (7) contains only two components, as the knowledge of the boundary and initial conditions has already been embedded in the state $\hat{\mathbf{u}}_{\theta_u}$.

For PINNs with re-parameterization, such PINNs are called hard-constrained PINNs, as those initial and boundary conditions are imposed by definition. On the other hand, for PINNs without re-parameterization, just like those in the original definition in the Related work 2, they are called soft-constrained PINNs, since their initial and boundary conditions are imposed as a loss function 'softly'. From here onwards, for simplicity, we use the abbreviation 'Hard-PINNs' to represent hard-constrained PINNs. Likewise, we will use 'Soft-PINNs' to represent soft-constrained PINNs.

### 3.3 Our Method: PDE-GAN

In this section, we will introduce how to integrate PINNs into the framework of GANs to solve PDEOC problems. Our method, PDE-GAN, innovatively combines the framework of GANs from Section 3.1 with the hard-constrained PINNs from Section 3.2. It adjusts the relationship between the PDE residual term and the cost objective term in solving PDEOC problems through the GANs framework. Unlike the line search method, which manually adjusts the weight $\omega$ to linearly balance the relationship between the two loss terms, the PDE-GAN method introduces two continuously updating discriminator networks that can nonlinearly adjust the relationship between the two loss terms in real time.

In previous sections, we denoted the system state with hard-constrained as $\hat{\mathbf{u}}_{\theta_u}$ and the control function as $\mathbf{c}_{\theta_c}$. To make the explanation clear, we will now use the classical notations from GANs. Hereafter, we use the generator symbols $G_u(x, t, \theta_u)$ $(G_u)$ and $G_c(x, t, \theta_c)$ $(G_c)$ to represent $\hat{\mathbf{u}}_{\theta_u}$ and $\mathbf{c}_{\theta_c}$, respectively, i.e.,

$$G_u(x, t, \theta_u) := \hat{\mathbf{u}}_{\theta_u}(x,t) = \Phi(\mathbf{u}_{\theta_u}(x,t)), \tag{8a}$$

$$G_c(x, t, \theta_c) := \mathbf{c}_{\theta_c}(x,t). \tag{8b}$$

Then define the "generated data" and "real data" in GANs. According to Equation (1), $LHS_u^{(i)}$ denote the PDE residual value at the nodes $\{(x_i, t_i)\}_{i=1}^{N_f}$. $LHS_c$ represents the cost objective value associated with the form of the optimal control problem (bolza and Lagrange-type problems). We set $RHS_u^{(i)} = a$ and $RHS_c = b$, which implies that we aim for the value of $LHS_u^{(i)}$ and $LHS_c$ to approach the target value $a, b$ as closely as possible during the update process of the trainable neural network parameters $\theta_u$ and $\theta_c$ (Generally, $a, b$ are set to zero, with their values depending on the specific problem.). The specific representations are as follows:

$$LHS_u^{(i)} := \mathcal{F}[G_u(x^{(i)}, t^{(i)}, \theta_u), G_c(x^{(i)}, t^{(i)}, \theta_c)], \tag{9a}$$

$$LHS_c := \mathcal{J}[G_u(x, t, \theta_u), G_c(x, t, \theta_c)], \tag{9b}$$

$$RHS_u^{(i)} := a, \tag{9c}$$

$$RHS_c := b. \tag{9d}$$

We use the symbol $D_u(y_1, \alpha_u)$ $(D_u)$ to denote the discriminator network monitoring the PDE residual term, where $y_1$ represents the $LHS_u^{(i)}$ or $RHS_u^{(i)}$, and $\alpha_u$ denotes its trainable

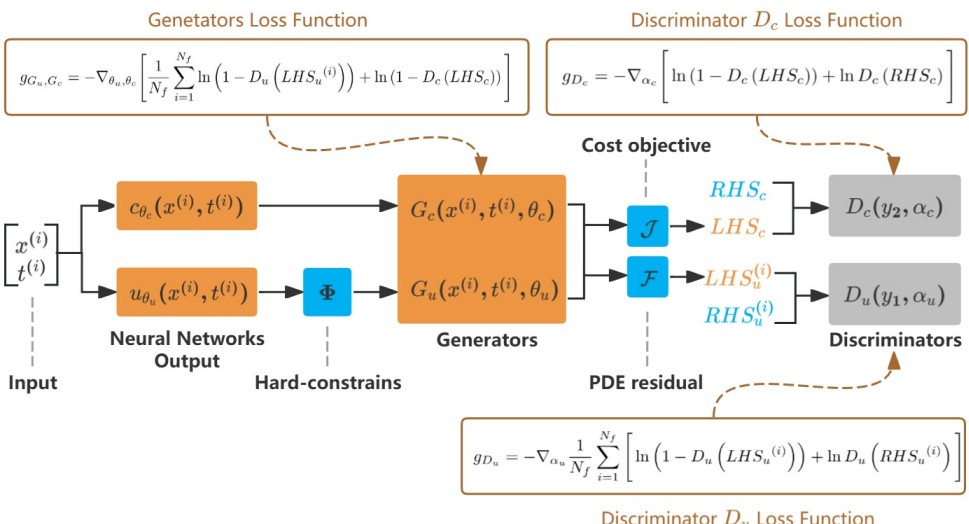

Figure 1: Schematic representation of PDE-GAN. We pass the input points $(x^{(i)}, t^{(i)})$ to two neural networks $u_{\theta_u}$ and $c_{\theta_c}$. Next, we analytically adjust $u_{\theta_u}$ using $\Phi$ to enforce hard constraint conditions (e.g., boundary and initial conditions), resulting in the generator networks $G_u$ and $G_c$. Automatic differentiation is applied to construct $LHS_u^{(i)}$ from the PDE residual $\mathcal{F}$. Subsequently, $LHS_u^{(i)}$ and $RHS_u^{(i)}$ are passed to the discriminator $D_u$, which is trained to evaluate whether $LHS_u^{(i)}$ is sufficiently close to $RHS_u^{(i)}$. After updating $D_u$, it provides new loss gradients to the generator for the PDE residual part ("forward"). Additionally, automatic differentiation is applied to construct $LHS_c$ from the cost objective $\mathcal{J}$. Then, $LHS_c$ and $RHS_c$ are passed to the discriminator $D_c$, which plays a similar role to $D_u$. After updating $D_c$, it provides new loss gradients to the generator for the cost objective part ("backward").

parameters. Similarly, $D_c(y_2, \alpha_c)$ $(D_c)$ represents the discriminator network monitoring the cost objective term, where $y_2$ is the $LHS_c$ or $RHS_c$, and $\alpha_c$ denotes its trainable parameters.

We update the trainable parameters of the generators $G_u$ and $G_c$ and the discriminators $D_u$ and $D_c$ according to the Binary Cross-Entropy loss 10, 11 and 12. Note that we perform stochastic gradient ascent for $G_u$ and $G_c$ (gradient steps $\propto g_{G_u, G_c}$), and stochastic gradient descent for $D_u$ and $D_c$ (gradient steps $\propto -g_{D_u}, -g_{D_c}$).

$$g_{G_u, G_c} = -\nabla_{\theta_u, \theta_c} \left[ \underbrace{\frac{1}{N_f} \sum_{i=1}^{N_f} \ln\left(1 - D_u\left(LHS_u^{(i)}\right)\right)}_{\text{forward}} + \underbrace{\ln\left(1 - D_c\left(LHS_c\right)\right)}_{\text{backward}} \right], \quad (10)$$

$$g_{D_u} = -\nabla_{\alpha_u} \frac{1}{N_f} \sum_{i=1}^{N_f} \left[ \ln\left(1 - D_u\left(LHS_u^{(i)}\right)\right) + \ln D_u\left(RHS_u^{(i)}\right) \right], \quad (11)$$

$$g_{D_c} = -\nabla_{\alpha_c} \left[ \ln\left(1 - D_c\left(LHS_c\right)\right) + \ln D_c\left(RHS_c\right) \right]. \quad (12)$$

The Equation (10) can be divided into two parts: we refer to the loss function representing the PDE residual part (The first part) as the "forward" loss and the loss function representing the cost objective part (The second part) as the "backward" loss. It can be seen that the gradients of $LHS_u^{(i)}$ and $LHS_c$ change with the variations of the discriminators $D_u$ and $D_c$. These changes adaptively adjust the gradient weights (for each node and cost objective),

which can be viewed as adjusting the relationship between the residuals at all training nodes and the cost objective at the node level. In contrast, in the Hard-PINNs method, the loss function (Equation (7)) keeps the ratio between the residuals and the cost objective for each training point fixed as $[1/N_f : 1/N_f : \cdots : 1/N_f : \omega]$, which is one of the reasons for the superior performance of our method. At the same time, our method continuously adjusts the relationship between the PDE residual and the cost objective in a nonlinear manner (Introduced by $D_u$ and $D_c$) within the GANs framework, providing greater flexibility. For complex problems (such as multi-scale phenomena), the optimization needs of different loss terms may change during training. Linear weights cannot adapt to this dynamic change in real-time, which may lead to some loss terms being over-optimized while others are neglected. The nonlinear approach (based on GAN-based adversarial learning) can dynamically adjust the optimization direction according to the current error distribution or the importance of the loss terms.

In line with the GANs training termination signal, we define $G1$, $D1$, $G2$ and $D2$ as follows:

$$G1 := -\frac{1}{N_f} \sum_{i=1}^{N_f} \ln\left(1 - D_u\left(LHS_u^{(i)}\right)\right), \tag{13}$$

$$D1 := -\frac{1}{2}\frac{1}{N_f} \sum_{i=1}^{N_f} \left[\ln\left(1 - D_u\left(LHS_u^{(i)}\right)\right) + \ln D_u\left(RHS_u^{(i)}\right)\right], \tag{14}$$

$$G2 := -\ln\left(1 - D_c\left(LHS_c^{(j)}\right)\right), \tag{15}$$

$$D2 := -\frac{1}{2}\left[\ln\left(1 - D_c\left(LHS_c^{(j)}\right)\right) + \ln D_c\left(RHS_c^{(j)}\right)\right]. \tag{16}$$

According to the description of the PDE-GAN method above, when the training is successful and the $LHS_u^{(i)}$ representing the PDE residual ($\mathcal{F}$) at node $\{x^{(i)}, t^{(i)}\}$ is sufficiently small, the discriminator $D_u$ finds it difficult to distinguish between the $RHS_u^{(i)}$ and $LHS_u^{(i)}$ . At this point, the output values of both $D_u(LHS_u^{(i)})$ and $D_u(RHS_u^{(i)})$ approach 0.5. The equations represented by $G1$ and $D1$ are equal. Therefore, in the subsequent PDE optimal control problems, we determine the success of the training based on whether $G1$, $D1$, $G2$ and $D2$ all converge to $\ln(2)$. This serves as our criterion for determining whether the PDE-GAN method has been successfully trained. Training on $G_u$ and $D_u$ stops when the absolute difference between $G1$ and $D1$ is smaller than $bound_1$ for a consecutive period of $N_s$ epochs; likewise for $G_c$ and $D_c$ with $G2$ and $D2$.

During the training process of the aforementioned GAN, we adopted the Two Time-Scale Update Rule method (Heusel et al., 2017) and Spectral Normalization (Miyato et al., 2018) method to make the GANs training more stable. To improve the sensitivity of GANs to hyperparameters under the Adam optimizer, we introduced Instance Noise (Arjovsky & Chintala, 2017) and Residual Monitoring (Bullwinkel et al., 2022). We provide a schematic representation of PDE-GAN in Figure 1 and detail the training steps in Algorithm 1.

Our Advantages: We shall emphasize that our proposed method does not require any line search (particular way of hyperparameter tuning), unlike the vanilla PINNs, which heavily depend on its two-step line search method to find the optimal weight ($\omega$). Therefore, our method is much more lightweight and efficient, especially in terms of shorter computation time, the evidence of which we will further demonstrate in the section of Experiment 4.

---

**Algorithm 1** PDE-GAN

---

Input: Partial differential equation $\mathcal{F}$, Boundary condition $\mathcal{B}$, Initial condition $\mathcal{I}$, Optimization objectives $\mathcal{J}$, generators $G_u(\cdot, \cdot; \theta_u)$ and $G_c(\cdot, \cdot; \theta_c)$, discriminators $D_u(\cdot; \alpha_u)$ and $D_c(\cdot; \alpha_c)$, grid $(x^{(i)}, t^{(i)})$ of $N_f$ points, re-parameterization function $\Phi$, total iterations $N$, stop signal bound $N_s$, $G_u$ and $G_c$ iterations $N_1$, $D_u$ iterations $N_2$, $D_c$ iterations $N_3$(without selecting iteration counts for the generators and discriminators, i.e., $N_1, N_2, N_3$=1), $Bound_u$, $Bound_c$.

Parameter: Learning rates $\eta_{G_u}, \eta_{G_c}, \eta_{D_u}, \eta_{D_c}$, Adam optimizer parameters $\beta_{G_u^1}, \beta_{G_u^2}, \beta_{G_c^1}, \beta_{G_c^2}, \beta_{D_u^1}, \beta_{D_u^2}, \beta_{D_c^1}, \beta_{D_c^2}$.

Output: $G_u$, $G_c$

    $S_u = 0$ and $S_c = 0$

    for $k = 1$ to $N$ do

      for $i = 1$ to $N_f$ do

        Forward pass $\mathbf{u}_{\theta_u} = \mathbf{u}_{\theta_u}(x^{(i)}, t^{(i)})$, $\mathbf{c}_{\theta_c} = \mathbf{c}_{\theta_c}(x^{(i)}, t^{(i)})$

        Analytic re-parameterization $G_u := \hat{u}_{\theta_u} = \Phi(u_{\theta_u})$,

        Compute $LHS_u^{(i)}$ (Equation 9a)

        Set $RHS_u^{(i)} = a$

      end for

      Compute $LHS_c$ (Equation 9b)

      Set $RHS_c = b$

      Compute gradients $g_{G_u}, g_{G_c}, g_{D_u}, g_{D_c}$ (Equation 10, 11 and 12)

      for $K_1 = 1$ to $N_1$ do

        Update generator $G_u$

        $\theta_u \leftarrow \text{Adam}(\theta_u, \eta_{G_u}, g_{G_u}, \beta_{G_u^1}, \beta_{G_u^2})$

        Update generator $G_c$

        $\theta_c \leftarrow \text{Adam}(\theta_c, \eta_{G_c}, g_{G_c}, \beta_{G_c^1}, \beta_{G_c^2})$

      end for

      for $K_2 = 1$ to $N_2$ do

        Update discriminator $D_u$

        $\alpha_u \leftarrow \text{Adam}(\alpha_u, -\eta_{D_u}, g_{D_u}, \beta_{D_u^1}, \beta_{D_u^2})$

      end for

      for $K_3 = 1$ to $N_3$ do

        Update discriminator $D_c$

        $\alpha_c \leftarrow \text{Adam}(\alpha_c, -\eta_{D_c}, g_{D_c}, \beta_{D_c^1}, \beta_{D_c^2})$

      end for

      if $|$ G1 - D1 $| <= Bound_1$ then

        $S_u = S_u + 1$

      else if  then

        $S_u = 0$

      end if

      if $|$ G2 - D2 $| <= Bound_2$ then

        $S_c = S_c + 1$

      else if  then

        $S_c = 0$

      end if

      if $S_u >= N_s$ and $S_c >= N_s$ then

        Break

      end if

    end for

    return $G_u$, $G_c$

---

## 4 Experiments

### 4.1 Experimental Setup and Evaluation Protocol

Benchmark Problems: We select several classic PDEOC problems, including both linear and nonlinear problems, as well as optimal control problems for boundary, spatio-temporal domain, and time-domain distributed equations. It is worth noting that, to verify the effectiveness of our method, we attempted the control function and cost objective in different scenarios: on the same boundary (Laplace problem), on opposite boundaries (Viscous Burgers initial value control problem), in the spatio-temporal domain (Viscous Burgers distributed control problem), and in the time-domain (Inviscid Burgers equation). On the four optimal problems, we test and compare the performance of (1) Soft-PINNs, (2) Hard-PINNs and (3) PDE-GAN respectively. More details of problems are listed in Appendix A.

(1) Laplace' Equation. The optimal boundary control problem of the Laplace equation is widely applied in various engineering and scientific fields, particularly in heat conduction, fluid mechanics, acoustics, and material design.

(2) Inviscid Burgers' Equation. The time-dependent distributed control problem for the inviscid Burgers' equation refers to adjusting control inputs over a given time interval to ensure that the system's state reaches a desired target in both time and space. Such problems are commonly used in the optimal control of dynamic systems and are relevant to fields such as fluid dynamics, traffic flow, and meteorological models.

(3) Viscous Burgers' Equation (Initial value control). The initial value control problem for the viscous Burgers' equation also has wide applications in fluid mechanics, traffic flow, meteorological simulation, and other fields. By optimizing and adjusting the system's initial state, it is possible to effectively control the subsequent evolution of the system to achieve desired physical or engineering goals.

(4) Viscous Burgers' Equation (Distributed control). The space-time-dependent distributed control problem for the viscous Burgers' Equation primarily involves adjusting the system in both time and space by optimizing control inputs to achieve effective control of fluid dynamic behavior.

Hyperparameters and Evaluation Protocols: For above problems, we construct the generator networks ($G_u$,$G_c$) and discriminator networks ($D_u$,$D_c$) using four multi-layer perceptrons (MLPs). We train these networks with the Adam optimizer (Diederik, 2014), where the learning rate decreases proportionally to the steps number by a factor of $\beta$. Since our top priority is on finding the optimal control for the problems, we apply high-precision numerical methods (Forward Euler Method, Finite Element Method and Spectral Method) to evaluate the trained optimal control $\mathbf{c}_\theta$ directly. The $\mathbf{u}_\theta$ will not be evaluated as it is only a side product of our training process. The cost objective ($\mathcal{J}$) obtained from numerical methods serves as our evaluation metric. In Soft-PINNs and Hard-PINNs, we simulated all results with weights ranging from 1e-03 to 1e11 (large cross-domain). In Appendix B, a comparative analysis of the three methods in different numerical experiments is presented. Additional details and method-specific hyperparameters (weights, neural network structures, learning rate, decay steps, decay rate, Adam optimizer parameters, activation function, and training termination criterias) are reported in Appendix C. The experiments are run on a single NVIDIA GeForce 4060 Ti GPU.

### 4.2 Main Results

The results of the four PDEOC problems are presented in Table 1. The data in the table represents the cost objective ($\mathcal{J}$) of three methods for different problems. A smaller value indicates better control performance. We bolden the best results of the four PDEOC problems. From the table, it can be seen that in all PDEOC problems, PDE-GAN achieved the lowest $\mathcal{J}$ than Soft-PINNs and Hard-PINNs without requiring line search.

Laplace: In the Laplace problem, the $\mathcal{J}$ value calculated by Soft-PINNs (1.01) is significantly larger than that of Hard-PINNs (7.57e-05) and PDE-GAN (1.13e-05). This indicates that when Soft-PINNs struggle to solve the problem, PDE-GAN can indeed enhance control performance. Experimental results demonstrate that hard-constraints help PINN to reduce

Table 1: PDEOC Problems Cost Objective

| Cost Objective ($\mathcal{J}$) | Laplace | Invis-Burgers | Vis-Burgers (Ini) | Vis-Burgers (Dis) |
|---|---|---|---|---|
| PINN-Soft | 1.01 | 7.74e-04 | 7.31e-05 | 2.43e-03 |
| PINN-Hard | 7.57e-05 | 1.04e-07 | 6.62e-06 | 1.54e-03 |
| Ours (PDE-GAN) | 1.13e-05 | 5.94e-09 | 2.32e-06 | 1.25e-03 |

Table 2: PDEOC Problems Running Time (Minute)

| Time (min) | Laplace | | Invis-Burgers | | Vis-Burgers (Ini) | | Vis-Burgers (Dis) | |
|---|---|---|---|---|---|---|---|---|
| | Mean | Total | Mean | Total | Mean | Total | Mean | Total |
| PINN-Soft | 2.9 | 43.7 | 1.0 | 15.3 | 3.5 | 52.7 | 1.62 | 24.3 |
| PINN-Hard | 3.6 | 54.6 | 1.5 | 23.3 | 4.2 | 63.15 | 1.7 | 25.4 |
| Ours (PDE-GAN) | 8.0 | | 5.1 | | 4.1 | | 3.3 | |

its $\mathcal{J}$ by around 4 orders of magnitude, while our method further reduces $\mathcal{J}$ by 7 times. Overall, our method achieves a $\mathcal{J}$ value that is about 5 orders of magnitude lower than that of the Soft-PINNs.

Invis-Burgers: In the Invis-Burgers problem, the $\mathcal{J}$ value calculated by Soft-PINNs (7.74e-04) is still significantly larger than that of Hard-PINNs (1.04e-07) and PDE-GAN (5.94e-09). Experimental results demonstrate that hard constraints can reduce the $\mathcal{J}$ of PINN by around 4 orders of magnitude, while our method further reduces $\mathcal{J}$ by 18 times. Overall, our method achieves a $\mathcal{J}$ value that is 5 orders of magnitude lower than that of the Soft-PINNs.

Vis-Burgers (Ini): In the Vis-Burgers initial value control problem, the $\mathcal{J}$ calculated by Hard-PINNs (6.62e-06) is reduced by 10 times compared to Soft-PINNs (7.31e-05). Our method further reduces the $\mathcal{J}$ value by 3 times. Overall, PDE-GAN (2.32e-06) achieves a $\mathcal{J}$ value that is 30 times lower than that of Soft-PINNs.

Vis-Burgers (Dis): In the Vis-Burgers distributed control problem, although the cost objectives obtained by the three methods are quite similar, PDE-GAN (1.25e-03) can directly find the distributed control that minimizes the $\mathcal{J}$ without the need for line search. This significantly saves computation time, further demonstrating the advantages of our method in both accuracy and efficiency. In the next section, we will demonstrate that the PDE-GAN method does not require line search by comparing the training times of the three methods across different problems, which can greatly save computation time and improve solution efficiency.

### 4.3 Running Time analysis

Table 2 presents the total training time for Soft-PINNs, Hard-PINNs, and our method, along with the mean training time under a single weight setting. Although the training time for PINN methods is shorter with a single weight, the line search process requires repeated experiments with multiple weights (from 1e-03 to 1e11), leading to increased complexity and time consumption. In contrast, our method does not require line search and can find a better optimal control than both Soft-PINNs and Hard-PINNs more quickly and conveniently with just a single round of adversarial training.

## 5 Conclusion

This paper introduces PDE-GAN, a novel deep learning method for solving PDEs optimal control problems. By embedding the PINN structure into the GAN framework, we use two additional discriminator networks to adaptively adjust the loss function, allowing for the adjustment of weights between different competing loss terms. Compared to Soft-PINNs and Hard-PINNs, PDE-GAN can find the optimal control without the need for cumbersome line search, offering a more flexible structure, higher efficiency, and greater accuracy.

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
