## A  DETAILS OF PDEOC PROBLEMS

### A.1  THE LAPLACE PROBLEM

Consider the Laplace equation in the unit square $\Omega$ with Dirichlet boundary conditions

$$\begin{cases} \frac{\partial^2 u}{\partial x^2} + \frac{\partial^2 u}{\partial y^2} = 0 & x \in [0,1],\ y \in [0,1], \\ u(x,1) = c(x) \\ u(x,0) = \sin \pi x \\ u(0,y) = u(1,y) = 0, \end{cases} \tag{17}$$

where **c** is the control applied to the top wall. We seek to solve the convex optimal control problem

$$c^* = \arg\min_c \mathcal{J}(u,c) \quad \text{subject to equation (17),} \tag{18}$$

with

$$\mathcal{J}(u,c) = \int_0^1 \left| \frac{\partial u}{\partial y}(x,1) - u_a(x,1) \right|^2 dx. \tag{19}$$

In other words, we want to find the optimal control solution $c^*(x)$ at the top wall that produces the desired flux $u_a(x,1) = -\frac{\pi}{2}\sin(\pi x)$. This problem has the analytical optimal solution

$$c^*(x) = \operatorname{sech}(\pi)\sin(\pi x) - \frac{1}{2}\tanh(\pi)\sin(\pi x), \tag{20}$$

corresponding to the state solution

$$u^*(x,y) = \frac{1}{2}\operatorname{sech}(\pi)\sin(\pi x)\left(e^{\pi(y-1)} + e^{\pi(1-y)}\right) - \frac{1}{4}\operatorname{sech}(\pi)\sin(\pi x)\left(e^{\pi y} - e^{-\pi y}\right). \tag{21}$$

The linearity of the Laplace equation implies that any PDEOC problems defined by a quadratic cost objective will be convex (Tröltzsch, 2010). Therefore, in Appendix B.1 we only need compare the optimal control $c(x)$ with analytical optimal solutin $c^*(x)$.

In Figure 2, we present the state solution(Fig2 a) and the system state under the control of the PDE-GAN method for this problem(Fig2 b). In this problem, we utilized the finite element method to compute the cost objective of the three methods, thereby evaluating the advantages and disadvantages of each approach.

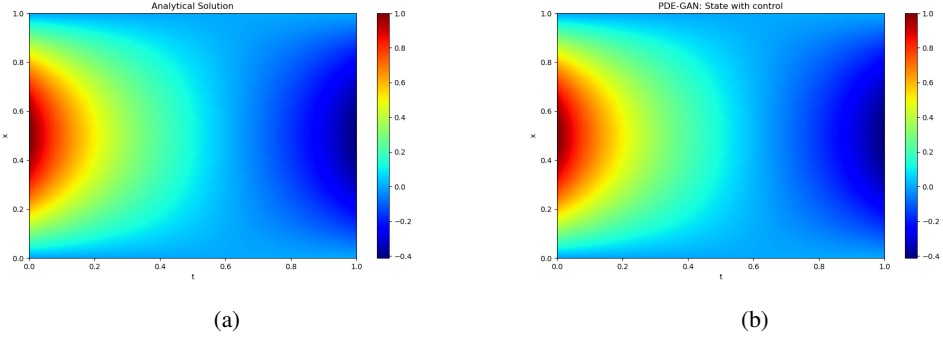

(a)          (b)

Figure 2: Laplace: (a) PDEOC problems analytical solution; (b) The system state under the control of the PDE-GAN method (Simulated using high fidelity Gauss-Seidel Iterative Method with Central Difference Scheme).

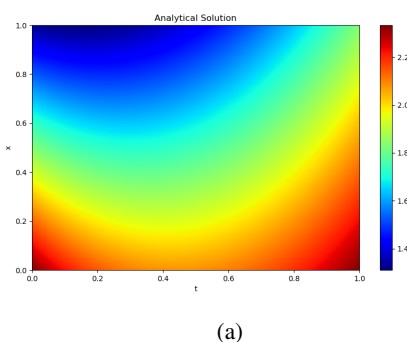 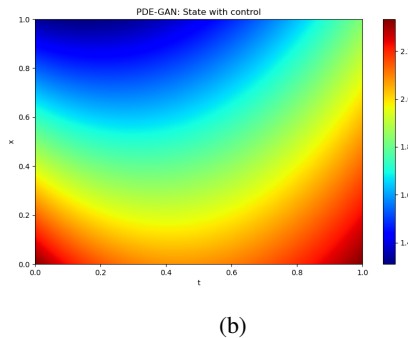

(a)                      (b)

Figure 3: Inviscid Burgers: (a) PDEOC problems analytical solution; (b) The system state under the control of the PDE-GAN method (Simulated using high fidelity Forward Euler Method (FEM) method).

## A.2 THE INVISCID BURGERS PROBLEM

We then consider the one-dimensional Inviscid-Burgers equation with Dirichlet boundary conditions, a prototypical nonlinear hyperbolic PDEs that takes the form

$$\begin{cases} \frac{\partial u}{\partial t} + u\frac{\partial u}{\partial x} = c(t) & x \in [0,1], \ t \in [0,1] \\ u(0,t) = \frac{1}{t+1} + \frac{(t+1)^2}{3} \\ u(1,t) = \frac{2}{t+1} + \frac{(t+1)^2}{3} \\ u(x,0) = x + 4/3, \end{cases} \tag{22}$$

where $c(t)$, the control signal, is a function $c(t) \in R$ depends on time $t$ but does not depend on spatial coordinates x. We seek to solve this optimal control problem

$$c^* = \arg\min_c \mathcal{J}(c) \quad \text{subject to equation (22)}, \tag{23}$$

with

$$\mathcal{J}(u,c) = \int_0^1 |u(x,1) - u_a(x,1)|^2 dx, \tag{24}$$

where

$$u_a(x,1) = \frac{x+1}{2} + \frac{4}{3}. \tag{25}$$

In other words, we seek the optimal control $c(x)$ that results in the same final state as equation 25. This problem has an analytical optimal control given by equation 26.

$$c^*(t) = t + 1, \tag{26}$$

corresponding to the state solution

$$u_a(x,t) = \frac{x+1}{t+1} + \frac{(t+1)^2}{3}. \tag{27}$$

To evaluate the quality of the optimal control found by the three methods, we use the Forward Euler Method to compute the final state $u(x,1)$ corresponding to the optimal initial condition $c^*(x)$, where the difference step sizes are $dx = 1e\text{-}3$, $dy = 5e\text{-}6$.

In Figure 3, we present the state solution(Fig3 a) and the system state under the control of the PDE-GAN method for this problem(Fig3 b).

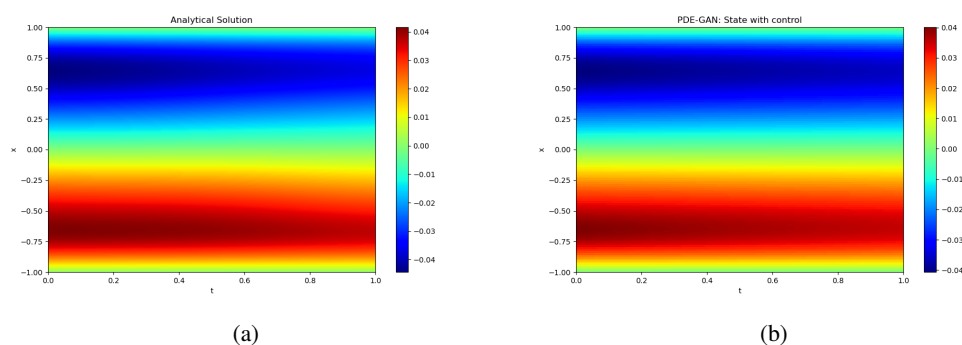

(a)                  (b)

Figure 4: Viscous Burgers (Initial value control): (a) PDEOC problems analytical solution; (b) The system state under the control of the PDE-GAN method (Simulated using high fidelity 128-order fourier spectral method).

## A.3 THE VISCOUS BURGERS PROBLEM (INITIAL VALUE CONTROL)

We then consider the one-dimensional Viscous Burgers equation with Dirichlet boundary conditions, a prototypical nonlinear hyperbolic PDEs that takes the form

$$
\begin{cases}
\frac{\partial u}{\partial t} + u\frac{\partial u}{\partial x} = v\frac{\partial^2 u}{\partial^2 x} & x \in [-1,1],\ t \in [0,1] \\
u(-1,t) = 0 \\
u(1,t) = 0 \\
u(x,0) = c(x),
\end{cases}
\tag{28}
$$

where $c(x)$ is the initial condition. We seek to solve the non-convex optimal control problem

$$
c^* = \arg\min_c \mathcal{J}(c) \quad \text{subject to equation (28),}
\tag{29}
$$

with

$$
\mathcal{J}(u,c) = \int_{-1}^{1} |u(x,1) - u_a(x,1)|^2 dx,
\tag{30}
$$

where

$$
u_a(x,1) = \frac{2\nu\pi \sin(\pi x)}{2 + \cos(\pi x)}.
\tag{31}
$$

In other words, we seek the optimal initial condition $c(x)$ that results in the same final state as equation (31). This problem has an analytical optimal initial condition given by equation 32.

$$
u_a(x,0) = \frac{2\nu\pi e^{\pi^2\nu} \sin(\pi x)}{2 + e^{\pi^2\nu} \cos(\pi x)},
\tag{32}
$$

corresponding to the state solution

$$
u_a(x,t) = \frac{2\nu\pi e^{-\pi^2\nu(t-1)} \sin(\pi x)}{2 + e^{-\pi^2\nu(t-1)} \cos(\pi x)}.
\tag{33}
$$

To evaluate the quality of the optimal solutions found by the three methods, we use a spectral solver to compute the final state $u(x,1)$ corresponding to the optimal initial condition $c^*(x)$ in the framework of the three methods, where we selected 128 equally spaced nodes. In Figure 4, we present the state solution(Fig4 a) and the system state under the control of the PDE-GAN method for this problem(Fig4 b).

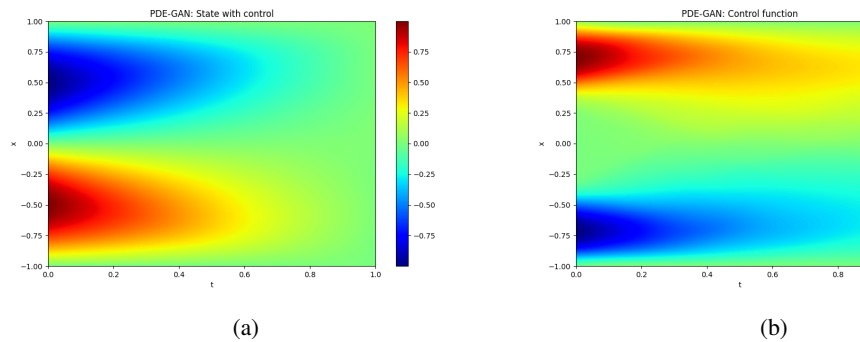

(a)                                   (b)

Figure 5: Viscous Burgers (Distributed control): (a) The system state under the control of the PDE-GAN method (Simulated using high fidelity Forward Euler Method (FEM) method); (b) PDE-GAN distributed optimal control $f(x, t)$.

### A.4 THE VISCOUS BURGERS PROBLEM (DISTRIBUTED CONTROL)

Finally, we consider the one-dimensional non-homogeneous Viscous Burgers equation with Dirichlet boundary conditions, which has the following form

$$\begin{cases} \frac{\partial u}{\partial t} + u\frac{\partial u}{\partial x} - v\frac{\partial^2 u}{\partial^2 x} = f(x,t) & x \in [-1, 1], \, t \in [0, 1] \\ u(-1, t) = 0 \\ u(1, t) = 0 \\ u(x, 0) = sin(\pi x), \end{cases} \quad (34)$$

where $f(x, t)$ is the distributed control. We seek to solve the optimal control problem

$$c^* = \arg\min_c \mathcal{J}(c) \quad \text{subject to equation (34)}, \quad (35)$$

with

$$\mathcal{J}(u, c) = \int_{-1}^{1} |u(x, 1)|^2 dx + \sigma \int_{-1}^{1} \int_0^1 |f(x, t)|^2 dt dx, \quad (36)$$

In other words, we seek the optimal distributed control $f(x, t)$ that minimizes the energy, such that the PDEs system trends toward the unstable zero solution at the final time. We select $\sigma$ as 0.001.

To evaluate the quality of the optimal solutions found by the three methods, we use Forward Euler Method to compute the cost objective corresponding to the optimal distributed control $f(x, t)$ in the framework of the three methods, where the difference step sizes of FEM are $dx$ = 4e-3, $dy$ = 2.5e-4.

We present the system state under the control of the PDE-GAN method for this problem(Fig5 a) and the distributed control of the PDE-GAN method (Fig5 b).

## B RESULTS ANALYSIS

In this section, we will analyze the advantages of the PDE-GAN method in four PDEOC problems. Moreover, we will take the Laplace equation as an example to explain in detail the symbolic definition and image implications of the PDE-GAN method that appear in all four problems.

### B.1 LAPLACE

In Figure 6 (a) both $G1$ and $D1$ converge to the same value $\ln(2)$, which means the $LHS_u$ representing the PDE residual ($\mathcal{F}$) is sufficiently small and the discriminator $D_u$ finds it difficult to distinguish between the $RHS_u$ ('$real_u$') and $LHS_u$ ('$fake_u$'). Similarly, in Figure 6 (b), when the $LHS_c$ representing the cost objective ($\mathcal{J}$) is sufficiently small, the outputs value of $D_c(LHS_c)$ and $D_c(RHS_c)$ also approach 0.5 and $G2$ and $D2$ converge to the same value $\ln(2)$.

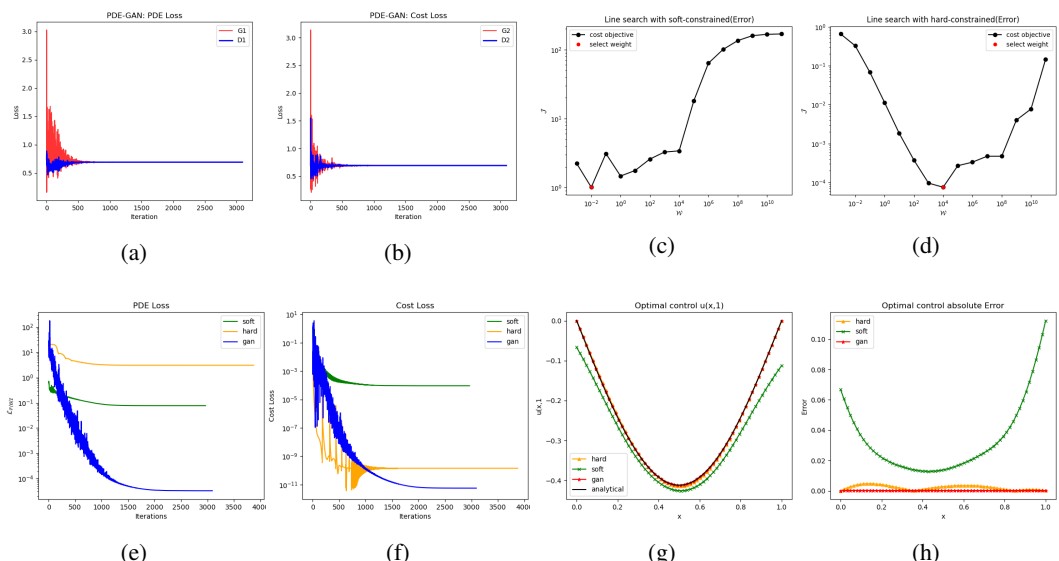

Figure 6: Laplace: (a) Convergence of the $G1$ and $D1$ in the PDE-GAN method; (b) Convergence of the $G2$ and $D2$ in the PDE-GAN method; (c) The cost objective of Soft-PINNs under different weights ($\omega_s$=1e-02). (d) The cost objective of Hard-PINNs under different weights ($\omega_h$=1e04); (e) Comparison of PDE loss for Soft-PINNs and Hard-PINNs under optimal weights with the PDE-GAN method; (f) Comparison of Cost loss for Soft-PINNs and Hard-PINNs under optimal weights with the PDE-GAN method; (g) The optimal controls obtained by the three methods are compared with the analytical solution; (h) The absolute errors between the results of the three methods and the analytical solution.

Figure 6(c)-(d) shows the cost objectives of Soft-PINNs and Hard-PINNs under different weights based on the finite element method. We marked the optimal weights (Soft: 1e-02, Hard: 1e04) in the figure with red dots. Next, we will compare the results of Soft-PINNs and Hard-PINNs under the optimal weights with our results to demonstrate the advantages of PDE-GAN.

In Figure 6(e), we compared the PDE loss during the training processes of the Soft-PINNs, Hard-PINNs (optimal weight) and our method (PDE-GAN). It is obvious that the PDE loss of PDE-GAN is approximately at least 5 orders of magnitude lower than that of the other methods. In Figure 6(f), we compared the Cost loss during the training processes of the three methods. Additionally, our method can achieve a reduction of 7 and 2 orders of magnitude compared to Soft-PINNs and Hard-PINNs. To facilitate reader understanding, we reiterate the formulation of the PDE loss and cost loss based on the loss functions in (3) and (7) as follows.

**For Soft-PINNs:**

$$PDE\ loss = \mathcal{L}_{\mathcal{F}}(\mathbf{u}_{\theta_u}, \mathbf{c}_{\theta_c}) + \mathcal{L}_{\mathcal{B}}(\mathbf{u}_{\theta_u}) + \mathcal{L}_{\mathcal{I}}(\mathbf{u}_{\theta_u}). \tag{37}$$

$$Cost\ loss = \mathcal{L}_{\mathcal{J}}(\mathbf{u}_{\theta_u}, \mathbf{c}_{\theta_c}). \tag{38}$$

**For Hard-PINNs and PDE-GAN:**

$$PDE\ loss = \mathcal{L}_{\mathcal{F}}(\hat{\mathbf{u}}_{\theta_u}, \mathbf{c}_{\theta_c}) \tag{39}$$

$$Cost\ loss = \mathcal{L}_{\mathcal{J}}(\hat{\mathbf{u}}_{\theta_u}, \mathbf{c}_{\theta_c}). \tag{40}$$

In Figure 6(g), we plot the optimal control obtained $\mathbf{c}_{\theta_c}$ from the three methods and analytical solution $c^*(x)$. To make the comparison more intuitive, in Figure 6(h), we subtract the analytical solution from each of the three solutions and take the absolute value. It can be observed that the PDE-GAN method achieves the highest accuracy without the need for line search.

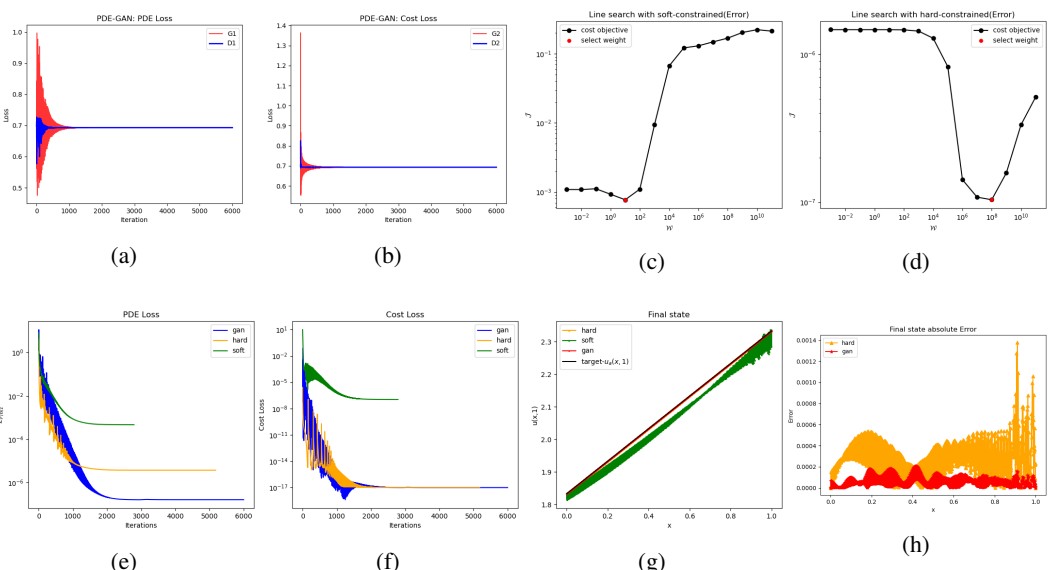

(a)  (b)  (c)  (d)

(e)  (f)  (g)  (h)

Figure 7: Inviscid Burgers: (a) Convergence of the $G1$ and $D1$ in the PDE-GAN method; (b) Convergence of the $G2$ and $D2$ in the PDE-GAN method; (c) The cost objective of Soft-PINNs under different weights ($\omega_s$=1e01). (d) The cost objective of Hard-PINNs under different weights ($\omega_h$=1e08); (e) Comparison of PDE loss for Soft-PINNs and Hard-PINNs under optimal weights with the PDE-GAN method; (f) Comparison of Cost loss for Soft-PINNs and Hard-PINNs under optimal weights with the PDE-GAN method; (g) Numerical solutions for the final state of Soft-PINNs, Hard-PINNs, and PDE-GAN, compared to the target final state; (h) Absolute value of the error between the final states of Hard-PINNs and PDE-GAN and the target final state.

## B.2 INVISCID BURGERS

In Figure 7(a)-(b), both ($G1$,$D1$) and ($G2$, $D2$) converge to ln(2). Similar to Laplace problem, this means that we have successfully trained a generator network that can deceive the discriminator.

Figure 7(c)-(d) illustrates the cost objectives of Soft-PINNs and Hard-PINNs under different weights. Next, we compare the results of Soft-PINNs and Hard-PINNs under the optimal weights (Soft: 1e01, Hard: 1e08) with our results (PDE-GAN).

In Figure 7(e)-(f), although our method shows little difference from the Hard-PINNs in the Cost loss, in terms of PDE loss, our method (PDE-GAN) is nearly 2 orders of magnitude lower than the Hard-PINNs.

In Figure 7(g), the final state of the system $u(x, 1)$ under three methods is displayed, along with our reference target state $u_a(x, 1)$. It is evident that the Soft-PINNs is the farthest from the target state. To make the comparison more intuitive, in Figure 7(h), we subtract the target state $u_a(x, 1)$ from the two final states (Hard and ours). It can be visually observed that our method is closer to $u_a(x, 1)$.

## B.3 VISCOUS BURGERS (INITIAL VALUE CONTROL)

In Figure 8(a)-(b), both ($G1$,$D1$) and ($G2$, $D2$) converge to ln(2).

In Figure 8(c)-(d), we calculated the cost objective of the initial conditions for Soft-PINNs and Hard-PINNs under different weights using the 128th-order spectral method and selected the optimal weights (Soft: 1e04, Hard: 1e05).

In Figure 8(e)-(f), we compared the PDE loss and Cost loss during the training processes of the Soft-PINNs, Hard-PINNs and our method (PDE-GAN). Although our method shows little difference from the Hard-PINNs in the PDE loss, in terms of Cost loss, our method (PDE-GAN) is nearly 2 orders of magnitude lower than the Hard-PINNs.

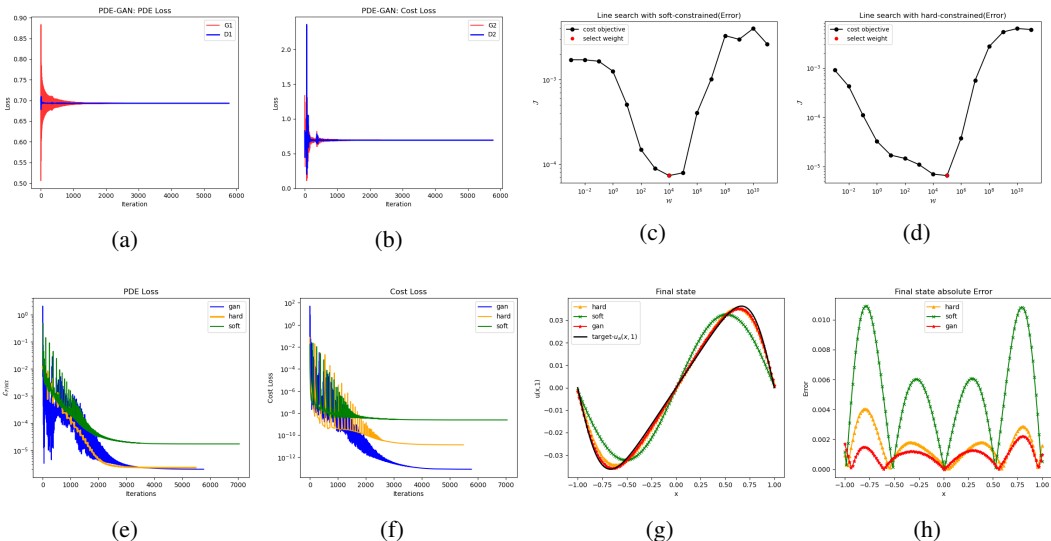

Figure 8: Viscous Burgers (Initial value control): (a) Convergence of the $G1$ and $D1$ in the PDE-GAN method; (b) Convergence of the $G2$ and $D2$ in the PDE-GAN method; (c) The cost objective of Soft-PINNs under different weights ($\omega_s$=1e04). (d) The cost objective of Hard-PINNs under different weights ($\omega_h$=1e05); (e) Comparison of PDE loss for Soft-PINNs and Hard-PINNs under optimal weights with the PDE-GAN method; (f) Comparison of Cost loss for Soft-PINNs and Hard-PINNs under optimal weights with the PDE-GAN method; (g) Numerical solutions for the final state of Soft-PINNs, Hard-PINNs, and PDE-GAN, compared to the target final state; (h) Absolute value of the error between the final states of Hard-PINNs and PDE-GAN and the target final state.

Figure 8(g) shows the spectral solution of the final state compared with the target state $u_a(x, 1)$. Additionally, to make the comparison more intuitive, in Figure 8(h), we present the absolute value of the difference between the spectral solution of the final state and the target final state $u_a(x, 1)$. It can be visually observed that our method is closer to the target.

### B.4 VISCOUS BURGERS (DISTRIBUTED CONTROL)

In Figure 9(a)-(b), both ($G1$,$D1$) and ($G2$, $D2$) converge to ln(2).

In Figure 9 (c)-(d), we calculated the cost objective of the optimal distributed control ($f(x,t)$) for Soft-PINNs and Hard-PINNs under different weights base on the Forward Euler Method.

In Figure 9 (e)-(f), we compared the changes in PDE loss and Cost loss during the training processes of the three methods. Although our method shows little difference in Cost loss compared to Hard-PINNs, our method (PDE-GAN) has a PDE loss at least two orders of magnitude lower than the other methods.

In Figure 9 (g), we show the numerical solutions (Forward Euler method) at the final state of three methods and compare them with the final target state $u_a(x, 1) = 0$. Additionally, to make the comparison more intuitive, in Figure 9 (h), we subtract the final state $u(x, 1)$ from $u_a(x, 1)$ and take the absolute value. It can be visually observed that our method is closer to the target state zero.

## C TRAINING DETAILS

In all optimal control problems, we compare the three methods under the same uniform grid conditions (32x32). Three methods all use generator $G_u$ to define the state solution($u(x, t)$) of the PDEOC problems, and generator $G_c$ to define the optimal control function($c(x)/f(x, t)/c(t)$). During training, we choose the Adam optimizer and simultaneously use a learning rate scheduler. $G_u$ and $G_c$ start with the same initial learning rate, and after every decay iterations, the learning rate is

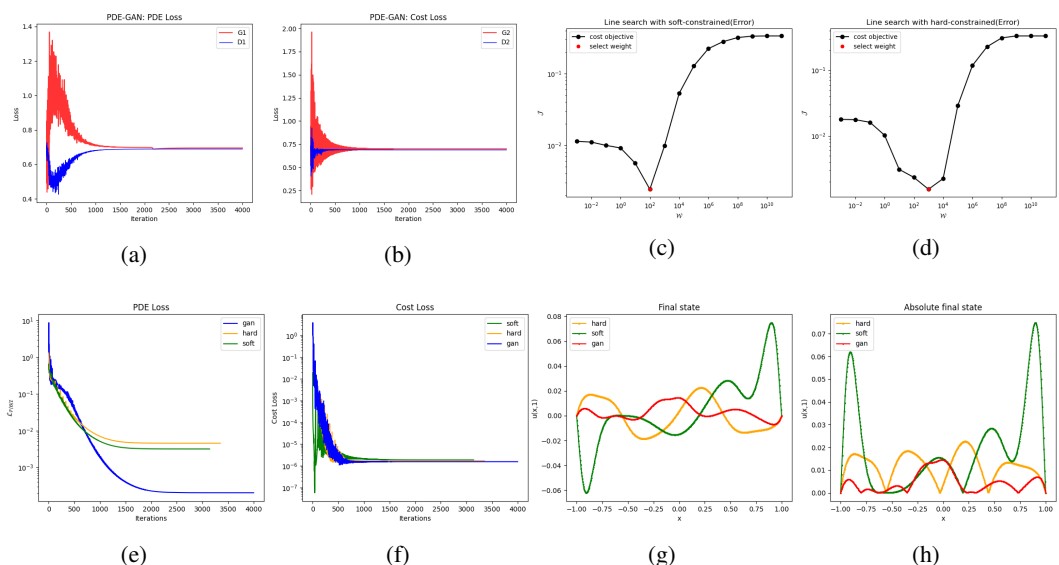

Figure 9: Viscous Burgers (Distributed control): (a) Convergence of the $G1$ and $D1$ in the PDE-GAN method; (b) Convergence of the $G2$ and $D2$ in the PDE-GAN method; (c) The cost objective of Soft-PINNs under different weights ($\omega_s$=1e02). (d) The cost objective of Hard-PINNs under different weights ($\omega_h$=1e03); (e) Comparison of PDE loss for Soft-PINNs and Hard-PINNs under optimal weights with the PDE-GAN method; (f) Comparison of Cost loss for Soft-PINNs and Hard-PINNs under optimal weights with the PDE-GAN method; (g) Numerical solutions for the final state of Soft-PINNs, Hard-PINNs, and PDE-GAN, compared to the target final state ($u_a(x, 1) = 0$); (h) Absolute value of the error between the final states of Hard-PINNs and PDE-GAN and the target final state.

multiplied by the decay factor $\gamma$. Also, to make the training process more stable, we added residual connections between the hidden layers of the neural network.

**PDE-GAN:** We reparameterize generator $G_u$ so that it directly satisfies the initial/boundary conditions of the problem. The structure of the generator($G_u, G_c$) and discriminator($D_u, D_c$) networks is shown in Table 3. We train both generator $G_u/G_c$ and discriminator $D_u/D_c$ simultaneously by loss function (10 - 12), starting from new parameter initializations. According to $G1$ (13), $D1$ (14), $G2$ (15) and $D2$ (16) mentioned before, we define the criteria for determining convergence. When $|G1 - D1|$ and $|G2 - D2|$ remain continuously below a certain upper bound for 500 iterations, we consider the training to be complete. This means that we generate data through generators $G_u$ and $G_c$ that are sufficient to deceive discriminators $D_u$ and $D_c$ continuously for 500 iterations. During the training process, the three loss functions (10 - 12) are updated in a 1:1:1 ratio sequentially. Additionally, the beta parameters of the Adam optimizer impact the training of generative adversarial networks. Therefore, we adopted the residual monitoring methodBullwinkel et al. (2022) to filter and select stable $\beta$ parameters for training. Tables 3 summarize these hyperparameter values for PDE-GAN.

**Soft-PINNs:** For Soft-PINNs, we train both generator $G_u/G_c$ simultaneously by loss function (3). The PINN method only utilizes the generators ($G_u, G_c$), and its hyperparameters are identical to those of the generators in the PDE-GAN method. We repeat above procedure for some values of $w_s$. We consider the result to have converged when the change in the PDEs residual term is continuously less than a upper bound for n times. The training hyperparameters for the Soft-PINNs are shown in Table 4.

**Hard-constrained:** For Hard-PINNs method, we reparameterize generator $G_u$ so that it directly satisfies the initial/boundary conditions of the problem. Then we train both generator $G_u/G_c$ simultaneously by loss function (7). Similar to Soft-PINNs, Hard-PINNs also only utilizes the generators

$(G_u, G_c)$, and its hyperparameters are identical to those of the generators in the PDE-GAN method. We repeat above procedure for some values of $w_h$. We consider the result to have converged when the change in the PDEs residual term is continuously less than a upper bound for n times. The training hyperparameters for the Hard-PINNs are shown in Table 5.

Table 3: PDE-GAN methods hyper-parameters.

| Hyperparameter | Laplace | Inviscid Burgers | BurgersV-ini | BurgersV-dis |
|---|---|---|---|---|
| $G_u$ | [2, (50*4), 1] | [2, (50*3), 1] | [2, (50*3), 1] | [2, (50*3), 1] |
| $G_c$ | [1, (50*4), 1] | [1, (50*3), 1] | [1, (50*3), 1] | [1, (50*3), 1] |
| $D_u$ | [1, (50*2), 1] | [1, (20*5), 1] | [1, (20*5), 1] | [1, (20*5), 1] |
| $D_c$ | [1, (50*2), 1] | [1, (20*5), 1] | [1, (20*5), 1] | [1, (20*5), 1] |
| Learning | $G_u/G_c$: 0.0121 | $G_u/G_c$: 0.0127 | $G_u/G_c$: 0.0127 | $G_u/G_c$: 0.0127 |
| rate(G/D) | $D_u/D_c$: 0.0884 | $G_u/G_c$: 0.0054 | $G_u/G_c$: 0.0054 | $G_u/G_c$: 0.0054 |
| Decay steps | 9 | 10 | 20 | 10 |
| Decay($\gamma$) | 0.9533 | 0.9548 | 0.9548 | 0.9548 |
| Grid | [32*32] | [32*32] | [32*32] | [32*32] |
| $G_u/G_c - \beta_1$(Adam) | 0.2955 | 0.1852 | 0.1852 | 0.1852 |
| $G_u/G_c - \beta_2$(Adam) | 0.3582 | 0.5941 | 0.5941 | 0.5941 |
| $D_u/D_c - \beta_1$(Adam) | 0.5751 | 0.0937 | 0.0937 | 0.0937 |
| $D_u/D_c - \beta_2$(Adam) | 0.1330 | 0.1846 | 0.1846 | 0.1846 |
| Activation | tanh | tanh | tanh | tanh |
| $Bound_u$ ($|G1 - D1|$) | 1e-4 | 1e-6 | 1e-6 | 1e-6 |
| $Bound_c$ ($|G2 - D2|$) | 1e-5 | 1e-6 | 1e-6 | 1e-6 |

Table 4: Soft-PINNs hyper-parameters.

| Hyperparameter | Laplace | Inviscid Burgers | BurgersV-ini | BurgersV-dis |
|---|---|---|---|---|
| Weight ($\omega_j$) | [1e-1 - 1e6] | [1e-3 - 1e5] | [1e-1 - 1e7] | [1e-3 - 1e5] |
| $G_u$ | [2, (50*4), 1] | [2, (50*3), 1] | [2, (50*3), 1] | [2, (50*3), 1] |
| $G_c$ | [1, (50*4), 1] | [1, (50*3), 1] | [1, (50*3), 1] | [1, (50*3), 1] |
| Learning | $G_u/G_c$: 0.0121 | $G_u/G_c$: 0.0127 | $G_u/G_c$: 0.0127 | $G_u/G_c$: 0.0127 |
| Decay steps | 9 | 10 | 20 | 10 |
| Decay($\gamma$) | 0.9533 | 0.9548 | 0.9548 | 0.9548 |
| Grid | [32*32] | [32*32] | [32*32] | [32*32] |
| $G_u/G_c - \beta_1$(Adam) | 0.2955 | 0.1852 | 0.1852 | 0.1852 |
| $G_u/G_c - \beta_2$(Adam) | 0.3582 | 0.5941 | 0.5941 | 0.5941 |
| Activation | tanh | tanh | tanh | tanh |
| Upper bound | 1e-9 | 1e-9 | 1e-12 | 1e-9 |
| Times (n) | 300 | 300 | 300 | 300 |

Table 5: Hard-PINNs hyper-parameters.

| Hyperparameter | Laplace | Inviscid Burgers | BurgersV-ini | BurgersV-dis |
|---|---|---|---|---|
| Weight ($\omega_j$) | [1 - 1e9] | [1e3 - 1e11] | [1e-1 - 1e7] | [1e-3 - 1e5] |
| $G_u$ | [2, (50*4), 1] | [2, (50*3), 1] | [2, (50*3), 1] | [2, (50*3), 1] |
| $G_c$ | [1, (50*4), 1] | [1, (50*3), 1] | [1, (50*3), 1] | [1, (50*3), 1] |
| Learning | $G_u/G_c$: 0.0121 | $G_u/G_c$: 0.0127 | $G_u/G_c$: 0.0127 | $G_u/G_c$: 0.0127 |
| Decay steps | 9 | 10 | 20 | 10 |
| Decay($\gamma$) | 0.9533 | 0.9548 | 0.9548 | 0.9548 |
| Grid | [32*32] | [32*32] | [32*32] | [32*32] |
| $G_u/G_c - \beta_1$(Adam) | 0.2955 | 0.1852 | 0.1852 | 0.1852 |
| $G_u/G_c - \beta_2$(Adam) | 0.3582 | 0.5941 | 0.5941 | 0.5941 |
| Activation | tanh | tanh | tanh | tanh |
| Upper bound | 1e-9 | 1e-15 | 1e-12 | 1e-9 |
| Times (n) | 300 | 300 | 300 | 300 |