# OpenReview forum: "PDE-GAN for solving PDE optimal control problems more accurately and efficiently"
_ICLR.cc/2025/Conference — Submitted to ICLR 2025_

### Official Review · Reviewer_tvho · 2024-10-26

**Soundness:** 2
**Presentation:** 3
**Contribution:** 2
**Rating:** 3
**Confidence:** 5

**Summary:**

The aim of this work is to use neural networks to solve PDE-constrained optimal control problems. The main contribution of this work is to introduce the GAN style to train the PINN to solve optimal control problems. The GAN style to train PINN is the previous work.

**Strengths:**

This paper is not difficult to follow. The proposed method uses the PINN framework to solve the parametric optimal control problems, which can be used to solve high-dimensional problems. The training style is inspired by GAN. Based on such training style, the different terms in the loss can be balanced without tuning by hand.

**Weaknesses:**

1. The PDE-constrained optimal control problems considered in this work only involve the equality constraint, but in practice, the inequality constraints are typical, e.g., the box constraint.

2. As stated above, if there exist inequality constraints, the proposed method in this manuscript cannot be applied directly. There are some literature that have already resolved this issue, but this manuscript did not mention, e.g., P. Yin, G. Xiao, K. Tang, and C. Yang, AONN: An adjoint neural network method for all-at-once solutions of parametric optimal control problems, SIAM Journal on Scientific Computing (2024). In this literature, the authors handle more general parametric optimal control problems with complex constraints. Since the AONN inherits the scheme of direct adjoint looping, it does not require tuning the penalty parameter. At the very least, the author should discuss AONN in related work because its key point has a strong correlation with this manuscript.

3. The experimental results are not so convincing. The loss behavior during training is not shown. Only the final error is reported. However, the training procedure of GAN is unstable. It is hard to say that the performance is better than the baseline.

**Questions:**

1. The symbol of weight $w$ in line 97 is not consistent with equation (3).
2. The main point of this work is to remove the hand-picking of the penalty parameter $w$. $w$ is just one hyperparameter, but the discriminator is a network, and it has a lot of hyperparameters, such as the depth and the width. Moreover, PDE-GAN (the proposed method in this manuscript) needs four discriminators. Tuning one hyperparameter is easier than many hyperparameters.
3. Also, did you try to set $w$ to be learned?
4. As I said above, this work only considers the equality constraints, but the inequality constraints are common in practice. How do you generalize the proposed approach to more complex constraints?
5. The numerical experiments did not show the stability of PDE-GAN. Can you show the whole results (e.g., the loss during training) of $D_c$ and $D_u$ to demonstrate the stability of PDE-GAN?

---

> ### Author Response · Authors · 2024-11-18
>
> "W" represents answering weaknesses, and "Q" represents answering questions.
>
> W1W2Q4:
> Thank you for your question. Our method can indeed solve inequality-constrained problems. The solution approach can refer to LuLu [1]. If h(u,c)<0 represents an inequality constraint, we can define L_h = 1_{h(u,c)>0} * h^2(u,c) to measure the degree to which the system violates the inequality constraint. This term can then be incorporated into the loss function for gradient descent updates. We also greatly appreciate you bringing the AONN method to our attention. After studying it, we find it to be an innovative approach. We will consider introducing this work in the extended version of the PDE-GAN method.
>
> W3Q5:
> We have provided experimental results at the end of the paper. Detailed experimental setups, convergence plots, and hyperparameter settings for all four problems are included in the appendix PDF. Please refer to it for further information.
>
> Q1:
> Thank you very much for your meticulous review. The current version does have some omissions in the notation. We will carefully check and correct these issues.
>
> Q2
> Thank you for your question regarding the design of the GAN architecture. For the optimal control problem, we introduced two additional discriminator networks. The depth, width, and update hyperparameters of these networks were not specifically tuned. In other words, they were designed merely to distinguish between 0 and 1. The hyperparameter settings for the discriminators are included in the appendix PDF and follow the guidelines outlined in reference [2].
>
> Q3:
> There are already some methods, such as [1], that explore making w learnable. Our method introduces the PINN framework into a GAN architecture to determine whether the PDE loss and cost objective are sufficiently small (i.e., indistinguishable from 0). This approach adaptively updates the loss function to provide gradient updates for the two generators. Unlike existing linear adjustment methods for balancing the PDE loss and cost objective, our approach achieves nonlinear adjustments, resulting in better performance. However, for other methods, apart from line search, we have not yet identified a comprehensive and fair benchmark for comparison. This remains an area for future exploration.
>
> [1]Lu L, Pestourie R, Yao W, et al. Physics-informed neural networks with hard constraints for inverse design[J]. SIAM Journal on Scientific Computing, 2021, 43(6): B1105-B1132.
> [2]Heusel M, Ramsauer H, Unterthiner T, et al. Gans trained by a two time-scale update rule converge to a local nash equilibrium[J]. Advances in neural information processing systems, 2017, 30.

---

> > ### Comment · Reviewer_tvho · 2024-11-19
> >
> > Q2 Thank you for your question regarding the design of the GAN architecture...
> >
> > I am not curious about the GAN architecture. $w$ is just just one hyperparameter, but the discriminator is a network, and it has many hyperparameters. The motivation of PDE-GAN is to remove the hand-picking of the hyperparameter $w$. However, PDE-GAN introduces a lot of other hyperparameters that are hand-picking.

---

> > > ### Author Response · Authors · 2024-11-21
> > >
> > > First, the motivation of PDE-GAN is not merely to remove the hand-picking of the hyperparameter $w$ but to remove trial-and-error adjustments without theoretical guidance, while introducing a more guided weight updating process compared to line search. By incorporating two real-time updated discriminators into the loss function, we can easily introduce nonlinearity. Using the output of the discriminators as gradients enables more flexible and accurate adjustments to the relationship between the PDE residual and the cost objective.
> > >
> > > Second, while the discriminator network does introduce additional update parameters, it is only designed to construct a classifier to distinguish between 0 and 1. Its hyperparameter settings only need to ensure convergence and do not have as significant an impact on the results as the weight w in line search methods. The hyperparameter settings are based on [1], which provides five conditions (A1–A5), including update rate, decay rate, activation functions, and hyperparameters related to the Adam optimizer. The paper theoretically demonstrates that under these conditions, generative adversarial networks (GANs) can achieve a Nash equilibrium (convergence).
> > >
> > > Additionally, in the numerical experiments, for both the time-domain control problem (second numerical example) and the spatiotemporal distributed control problem (fourth numerical example), we used the exact same hyperparameter settings for the discriminator network. Our method achieved the best results in both cases. Using identical hyperparameter settings for different problems can be considered an ablation study to evaluate the sensitivity of different problems to hyperparameter choices.

---

### Official Review · Reviewer_ZZWP · 2024-10-27

**Soundness:** 2
**Presentation:** 1
**Contribution:** 2
**Rating:** 3
**Confidence:** 3

**Summary:**

The paper presents a GAN-based approach to address the dual optimization problem for solving PDEs with an unknown control function. The method integrates PINNs-like objective functions and loss structures, targeting both forward and inverse problems. In this setup, the generators are tasked with predicting both the control function and the corresponding solution function. Meanwhile, the discriminators are designed to differentiate between valid solutions and zero-valued outputs.

**Strengths:**

- Originality: The use of GANs under the framework of PINNs is interesting.
- Significance: The problem tackled in this paper is inherently challenging due to the complexity of solving inverse problems under strict physical constraints. The authors’ approach demonstrates a promising direction in addressing these difficulties effectively.

**Weaknesses:**

- Choice of the Discriminator:
  - The current approach computes discriminators in a point-by-point manner. However, in traditional settings with discrete images, the entire image is typically used as input instead of individual pixels. The authors should provide a clear rationale and experimental results for this design choice.

- Lack of Comprehensive Baseline Comparison:
  - The paper lacks a comparative analysis with relevant methods such as bi-level optimization techniques. While these methods are mentioned in the related work, the absence of a thorough experimental comparison is not adequately justified.
  - Furthermore, there is no comparison with existing approaches like Physics-informed DeepONet (Wang et al., 2021), which address similar challenges. A direct comparison would help contextualize the proposed method’s performance in relation to established approaches exploring similar ideas.

- Complexity of Addressed Problems:
  - The paper does not sufficiently communicate the complexity or importance of the problems it addresses, making it challenging for readers to assess the novelty and significance of the proposed solution.
  - For example, Mowlavi and Nabi (2022) explore a range of equations, from simpler Laplace problems to more complex 2D Navier-Stokes equations, in their study of PDE-based optimal control (PDEOC). Including results for similarly challenging equations in this work would strengthen the paper’s validation and impact.

- Readability and Clarity:
  - The submission requires revisions to enhance readability and clearly communicate the main ideas. Key areas for improvement include:
    - Unifying the notations for the generator, solution function, and control function.
    - Organizing and presenting the definitions of different components in a clearer and more cohesive manner.

References:
- Mowlavi and Nabi. (2022). *Optimal control of PDEs using physics-informed neural networks.*
- Wang et al. (2021). *Learning the solution operator of parametric partial differential equations with physics-informed DeepONets.*

**Questions:**

Please refer to Weaknesses.

---

> ### Author Response · Authors · 2024-11-18
>
> "W" represents answering weaknesses, and "Q" represents answering questions.
>
> W1:
> Thank you for your insightful question. Indeed, during training, we experimented with using both the entire image and single pixels as inputs. When training the discriminator Du, we required each node to satisfy the PDE residual conditions. Therefore, we opted for a discrete, point-by-point evaluation for Du. In contrast, when training the discriminator Dc, since the cost objective typically appears as an integral, we used the entire integral (image) as input. You can interpret our setup as follows: Nf > 1 (e.g., 32×32), and NT = 1.
>
> W2:
> We greatly appreciate your feedback regarding the lack of comparisons with other approaches (e.g., bilevel optimization methods, physics-informed DeepONets, etc.). Each method has its own scope of applicability and hyperparameter configurations that enhance its performance. We have not yet identified a unified setup that would allow for fair and comprehensive comparisons among them. However, we plan to explore how to achieve this in future work. In our comparative experiments, we ensured fairness by using the same optimizer, optimization parameters, and grid discretization method for all generators, differing only in the construction of the loss functions.
>
> W3:
> Thank you for pointing out this limitation. To illustrate the complexity of the control problems we tackled, we considered various control types:
> Control functions and cost objectives on the same boundary (e.g., Laplace problem),
> On opposite boundaries (e.g., viscous Burgers' initial value control problem),
> In the time domain only (e.g., viscous Burgers' distributed control problem),
> In the spatiotemporal domain (e.g., inviscid Burgers' equation).
>
> W4:
> We greatly value your critique of the paper’s writing and structure. We will reorganize these sections to improve clarity. Specifically, we will refine the background introduction and notation definitions and adjust the structure to make the content more coherent and accessible. Thank you again for your constructive feedback!

---

### Official Review · Reviewer_SHGp · 2024-10-31

**Soundness:** 1
**Presentation:** 1
**Contribution:** 2
**Rating:** 3
**Confidence:** 4

**Summary:**

This paper introduces PDE-GAN a framework for solving PDEs optimal control problem with a PINN and an adversarial loss. The framework is an extension of the hard-constraint PINNs, which imposes constraints directly on the PINN solution instead of enforcing them  through loss penalties. In the optimal control configuration, the pde and cost objectives are balanced by a weight $w$. Existing implementations require for a search of the best $w$ to find a compromise between the two loss terms. The adversarial loss aims at mitigating the need for searching for the optimal weight value. The authors conducted experiments on Laplace and Burgers equation with several control setups.

**Strengths:**

* The method seems to work and obtains good experimental results on the different problems.
* The overall running time is less than of the Soft-PINNs and Hard-PINNs baselines when linear search is taken into account.

**Weaknesses:**

* The motivations of the paper are not well-founded to my view. The authors do not explain why the adversarial approach is needed to balance the different loss terms and never discuss nor test possible alternatives.
* The paper has some writing issues and suffers from a lack of clarity. Sections 3.1 and 3.2 should be within a separate background section. The notations introduced in Section 3.3 are difficult to read, especially RHS and LHS which are not explicitly detailed. I suggest using several examples to improve clarity.
* The running time is greater than that of a single PINNs.
* The importance of linear search for the other methods is not explained properly.
* The results are only marginally better than Hard-PINNs except for the second equation.
* The authors do not discuss their architectural choices, especially the adversarial loss and the noise injection.
* I suggest changing the name of the framework to Adversarial-PINNs as it is more faithful to the core idea fo the paper.

**Questions:**

* Why would the adversarial loss help for solving PDE optimal control problems ?
* Have the authors tried other techniques to try balancing the two losses ?
* Have the authors tried without the hard constraints ?

---

> ### Author Response · Authors · 2024-11-18
>
> "W" represents answering weaknesses, and "Q" represents answering questions.
>
> W1：
> Thank you for raising this question!Unlike traditional PINN methods that adjust weight w(linearly balancing the PDE residual and cost objective), our approach continuously and nonlinearly adjusts the relationship between the PDE residual and the cost objective in the GAN framework by updating Du and Dc. This allows for greater flexibility.For complex problems (e.g., multi-scale phenomena), the optimization requirements of different loss terms may change during training. Linear weights cannot dynamically adapt to these changes, potentially leading to over-optimization of some loss terms while others are neglected. In contrast, the nonlinear approach based on GAN-based adversarial learning can dynamically adjust the optimization direction according to the current error distribution or the importance of each loss term.
>
> W2：
> In Section 3.1, we present the core GAN framework, while Section 3.2 focuses on the hard boundary constraint method. While these sections are not directly aimed at solving the PDEOC problem (background), they form integral components of our PDE-GAN approach, which is why we have structured the paper in this manner. Regarding Section 3.3, we will carefully check the notation. Specifically, the symbols LHSu and LHSc, as defined in Equations 8 and 9, respectively, represent the PDE residual and cost objective. Taking the Laplace equation problem as an example:LHSu corresponds to all the constraints in Equation 17, LHSc refers to the integral value in Equation 19, and both RHSu and RHSc are equal to 0.
>
> W3:
> For traditional line-search-based PINN methods, it is not possible to determine the optimal weights in a single training iteration. Although individual training runs are faster, the control performance is significantly worse. In contrast, our method eliminates the need for line search and achieves better control results than the PINN methods (both soft and hard).
>
> W4:
> In 2021, Mowlavi and Nabi proposed a more flexible framework than the traditional adjoint methods — a line-search PINN framework for optimal control — connecting optimal control problems with the deep learning community. Our research can be seen as an extension of the original PINN method, focusing on introducing nonlinear search methods to develop a more efficient and effective framework.
>
> W5:
> By employing hard constraints, we transformed the traditional soft constraint approach with four competing loss terms into a setup with only two competing loss terms. This allows for the use of dense line searches to find a reasonably good solution. However, our method surpasses this by avoiding the need for extensive trial-and-error searches, directly obtaining solutions better than those achieved with hard constraints.
>
> W6:
> The adversarial loss is introduced to implement a nonlinear adaptive weight search strategy. For details, please refer to Answer 1. Regarding noise injection, it is a classic technique in GAN training that helps improve model robustness. By adding noise, we introduce randomness during the training process, encouraging the network to explore a broader solution space and reducing reliance on specific data patterns. This enhances the model's performance under diverse input conditions.
>
> W7:
> Thank you for your suggestion. We named our method PDE-GAN because we aim to view the combination of GAN and PINN architectures as a novel approach to solving PDEOC problems. The main focus of our paper is to embed PINN into the GAN framework to enhance its capability of handling multiple loss terms, thereby improving its effectiveness in solving optimal control problems.
>
> Q1:
> Similar to the motivation we presented earlier for the PDE-GAN method, we chose to use Generative Adversarial Networks (GAN) as an optimization framework with adaptively changing loss functions. The key advantage of our method lies in its ability to continuously and nonlinearly adjust the proportions of different objectives through the updates of two discriminators. Compared to fixed-proportion updates (as used in soft and hard PINN approaches), our method is better suited for solving PDE optimal control problems.
>
> Q2:
> We are indeed interested in exploring other balancing techniques.However, each method has its own set of hyperparameters that can enhance its performance. At this stage, we have not identified a unified configuration that allows for a fair and comprehensive comparison. We will explore how to achieve such comparisons in future research.
>
> Q3:
> Thank you for your question. At present, we have not conducted experiments without hard constraints. The primary reason is that traditional PINN loss functions include the PDE residual term, boundary condition term, initial condition term, and cost objective term. These four losses often conflict with each other during optimization, which is a major factor contributing to the poor performance of traditional PINN methods.

---

> > ### Comment · Reviewer_SHGp · 2024-11-26
> > **Response**
> >
> > Thank you for your response.
> >
> > While I appreciate your reply, my concerns remain largely unaddressed due to the lack of additional results. As such, I am unable to increase my score.

---

### Official Review · Reviewer_az6D · 2024-11-01

**Soundness:** 3
**Presentation:** 1
**Contribution:** 2
**Rating:** 5
**Confidence:** 4

**Summary:**

This paper proposes to combine PDE constraints with generative adversarial training as a method to solve PDE optimal control problems. The method is a GAN-based analogue to PINNs and outperforms the latter in some numerical experiments.

**Strengths:**

The paper proposes a new method for PDE-constrained control problems and demonstrates its superiority to PINNs in several numerical experiments.

**Weaknesses:**

The paper is far from well-written. First, it contains many grammatical and spelling errors that distract from the overall contribution. Beyond this, the writing (especially in Section 3) is unclear and it is difficult to understand the authors' reasoning.

In addition, this paper lacks any comparison to classical techniques for solving PDE-constrained optimal control problems. The proposed method is only compared to PINNs, but PINNs are not exactly state-of-the-art methods and can be quite easy to beat in many circumstances. As such, I am not convinced that PDE-GAN is the best method for solving these problems.

**Questions:**

1. Why do we need two generators and two discriminators, what are their respective purposes?
2. What unit is used in Table 2? That is, are results presented in wallclock time?  Such information is relevant for anyone to make a fair comparison to this work.
3. Have the authors considered comparing their algorithm to classical techniques (instead of only PINNs)?

---

> ### Author Response · Authors · 2024-11-18
>
> "W" represents answering weaknesses, and "Q" represents answering questions.
> W1:
> Thank you for pointing out the errors. We will carefully review and improve the grammar and structure in the revised version to enhance the language quality of the paper, especially in Section 3. We also appreciate your feedback regarding the lack of comparisons with classical methods.
> For other methods, each has its specific application scope and hyperparameter settings to enhance performance. Currently, we have not identified a unified setting for a fair and comprehensive comparison of these methods. However, we will explore and study how to achieve such a comparison under a universally fair setting in future work.
> We can ensure that, in our comparative experiments, the same optimizer, optimization parameters, and mesh discretization methods were chosen for each generator. The only difference lies in the construction of the loss function, which makes the comparison more fair and comprehensive.
>
> Q1：
> Our goal is to address the imbalance between the optimization of the PDE residual term and the cost objective term when solving optimal control problems using the PINN method. To achieve this, we use two generators, Gu and Gc, to generate the surrogate model u of the system and the control function c, respectively. Additionally, we employ a discriminator Du to evaluate whether the loss of the PDE residual term is sufficiently close to zero, and another discriminator Dc to evaluate whether the loss of the cost objective term is sufficiently close to zero.During the adversarial training process, we iteratively update Du and Dc to nonlinearly adjust the relationships between the PDE residual and the cost objective terms. Compared to the traditional PINN approach, which requires manually adjusting the weights linearly, our method introduces a nonlinear adaptive update mechanism, offering greater flexibility.
>
> Q2:
> In the top-left corner of Table 2, we used "min" as the abbreviation for minutes. This time includes the GAN training time, and we ensured that the time for all three methods was calculated under the same settings, making the comparison valid.
>
> Q3：
> We fully recognize the importance of comparing with classical methods (such as the adjoint method) to evaluate the performance of new algorithms. However, each method has its own scope of applicability and hyperparameter settings that enhance its effectiveness. We have not yet identified a unified setup to ensure a fair and comprehensive comparison among them. Nevertheless, we will explore and investigate how to achieve such a comparison under a comprehensive and fair setting in future work.

---

> > ### Comment · Reviewer_az6D · 2024-12-03
> >
> > Thank you for clarifying the writing and improving the readability of your paper. This does strengthen the paper as compared to my initial reading.
> >
> > That being said, my main concern remains, i.e., the paper lacks a suitable comparison against traditional methods for PDEOC (such as the adjoint method discussed in Section 2 or bi-level optimization techniques as pointed our by another reviewer). Please note that I emphasize a more comprehensive numerical comparison precisely because this paper does not seem theory-oriented. While PDE-GAN is introduced, there is a lack of theoretical analysis in this work. In such a case, a more comprehensive numerical analysis seems to be the only way for the paper to produce sufficient contributions.
> >
> > I understand that it might not be easy to implement the adjoint method in exactly the same setup and it also requires more work on the user’s end, but it still would be interesting to see how well PDE-GAN compares against a naive implementation of the adjoint method. For instance, it would be useful to find a problem where the adjoint method blows up (or is computationally intractable), but PDE-GAN solves it seamlessly.
> >
> > Given the aforementioned, I would like to maintain my original rating.

---

### Official Review · Reviewer_mh1W · 2024-11-01

**Soundness:** 2
**Presentation:** 2
**Contribution:** 3
**Rating:** 5
**Confidence:** 4

**Summary:**

The paper proposes a novel method PDE-GAN, which integrates PINNs into the GANs framework to solve the PDEOC problems. The authors address the limitations of traditional PINN approaches in balancing competing loss terms and reducing computational time, particularly by eliminating the need for exhaustive line search in weight tuning. They validate their method on four representative PDEOC problems, including linear and nonlinear PDEs, and various types of control (boundary, spatio-temporal domain, and time-domain distributed equations) and compared with soft-PINNs and hard-PINNs.

**Strengths:**

The integration of PINNs into the GAN framework is a new approach for solving PDEOC problems. This allows to use two additional
discriminator networks to adaptively adjust the loss function, allowing for the adjustment of weights between different competing loss terms. Compared to Soft-PINNs and Hard-PINNs, PDE-GAN can find the optimal control without the need for cumbersome line search, offering a more flexible structure, higher efficiency, and greater accuracy.

**Weaknesses:**

The paper lacks a theoretical analysis explaining why integrating PINNs into a GAN framework results in improved performance. Theoretical insights or proofs would strengthen the paper, espeically without any line search, the comprehensive evaluations of the results could be beneficial, however, using the experimental results to address its advantages is the main weakness.

**Questions:**

In Algorithm 1, why do you just limit the number of epochs 500?

It seems that the algorithm updates the generator and discriminator together without any condition, why?

How do you properly set Bound1 and Bound2?

Table 2 shows the running time for PDE-GAN, which is the total? the mean? Does it include the training time for GAN?

---

> ### Author Response · Authors · 2024-11-18
>
> "W" represents answering weaknesses, and "Q" represents answering questions.
> W1:
> Thank you for raising this question! In the Introduction, lines 92–107, 110–111, and 282–291, we explained why we chose to integrate PINN into the GAN framework and how this approach enhances effectiveness.
> In our method, the two discriminators Du and Dc are continuously updated during training. According to the binary cross-entropy loss in the GAN framework (Eqs. 10, 11, and 12), the entire loss function (both parts) is dynamically adjusted, providing more accurate update gradients for the generators Gu and Gc.The reason is that unlike traditional PINN methods that adjust weight w(linearly balancing the PDE residual and cost objective), our approach continuously and nonlinearly adjusts the relationship between the PDE residual and the cost objective in the GAN framework by updating Du and Dc. This allows for greater flexibility.For complex problems (e.g., multi-scale phenomena), the optimization requirements of different loss terms may change during training. Linear weights cannot dynamically adapt to these changes, potentially leading to over-optimization of some loss terms while others are neglected. In contrast, the nonlinear approach based on GAN-based adversarial learning can dynamically adjust the optimization direction according to the current error distribution or the importance of each loss term.
>
> Q1：
> Thank you for your question. The number of epochs can be adjusted based on user requirements and does not need to be set specifically. In fact, during our experiments, we observed that the number of iterations for the PDE-GAN method typically ranged between 3500 and 6500. Therefore, we chose one-tenth of the mean of these values. If the differences between the two generators and discriminators remain smaller than Bound1 and Bound2 for 500 consecutive epochs, we consider the training process complete.
>
> Q2:
> Our algorithm updates the generators and discriminators based on Equations 10, 11, and 12 (the binary cross-entropy loss of generative adversarial networks). We adopted the traditional training method for GANs, “GANs Trained by a Two Time-Scale Update Rule,” as referenced in [1]. Following this method, the generator and discriminator are updated alternately (one full cycle in order), enabling the adversarial system to generate both the system state and the control function effectively.
> [1]Heusel, M., Ramsauer, H., Unterthiner, T., Nessler, B.,Klambauer, G., and Hochreiter, S. Gans trained by a two time-scale update rule converge to a nash equilibrium. CoRR, abs/1706.08500, 2017. URL http://arxiv.org/abs/1706.08500.
>
> Q3：
> The settings for Bound1 and Bound2 depend on the user's accuracy requirements. Please refer to lines 305-314 and the loss functions (Equations 10, 11, and 12). Taking Bound1 as an example, its error bound is defined as:   e^{-\text{Bound1}} < \frac{D_u(\text{LHS})}{1 - D_u(\text{RHS})} < e^{\text{Bound1}},and similarly for Bound2.
>
> Q4:
> Regarding the time aspect, please refer to lines 470-477. Here, we mention that, unlike the PINN method, which requires multiple training runs under different weight parameters www, the PDE-GAN method does not require line search; instead, it only needs a single round of adversarial training. Therefore, the runtime for PDE-GAN in Table 2 is the total runtime, without averaging, as only one training session is needed.
> This runtime includes the GAN training time, and we ensured that the time for all three methods was calculated under the same settings, making the comparison valid.

---

### Meta-Review · Area_Chair_anjw · 2024-12-20

**Metareview:**

This paper addresses PDE optimal control (PDEOC) problems, which optimize physical systems governed by partial differential equations (PDEs). While physics-informed neural networks (PINNs) are a recent approach, they struggle to balance competing loss terms. The authors propose PDE-GAN, a novel framework based on generative adversarial networks (GANs) to effectively manage these trade-offs.
Experiments on four representative PDEOC problems show that PDE-GAN achieves higher accuracy and faster computation times than PINNs, eliminating the need for line search.

The reviewers raised the following pros and cons:

Pros:
+ The integration of PINNs into a GAN framework (PDE-GAN) is innovative and offers a new approach to balancing competing loss terms in PDE optimal control problems.

+ PDE-GAN eliminates the need for line search, providing higher accuracy and reduced computational time compared to Soft-PINNs and Hard-PINNs.

+ The adversarial loss mechanism allows nonlinear and adaptive updates, improving performance on complex, multi-scale problems.

+ Experimental results demonstrate improvements over baseline PINNs in several numerical problems.

Cons:

- Lack of Theoretical Analysis: The paper lacks a solid theoretical explanation for why the GAN framework improves PINNs' performance, leaving the results largely empirically driven.

- Baseline Comparisons: The paper does not compare PDE-GAN with classical PDE optimal control methods, such as adjoint methods or bi-level optimization, making it harder to assess its true impact.

- Writing and Clarity: The paper contains grammatical errors, unclear sections, and inconsistent notation, particularly in the methods and results sections, affecting its readability.

- Limited Problem Scope: It focuses only on equality-constrained problems, whereas inequality constraints are more common in practice.

- GAN Stability: Concerns about the stability of GAN training are not addressed, as loss behaviors during training are not shown.

- Additional Hyperparameters: While PDE-GAN removes manual weight tuning, it introduces several new hyperparameters (e.g., discriminator settings), raising concerns about added complexity.

Despite the rebuttal addressing some weaknesses, reviewers maintained concerns about theoretical gaps, limited baseline comparisons, and overall scope. As a result the paper can not be accepted at this time.

**Additional Comments On Reviewer Discussion:**

The authors rebuttal did not seem to affect the reviewers assessment of the paper significantly

---

### Decision · Program_Chairs · 2025-01-22

Reject